# The cristae modulator Optic atrophy 1 requires mitochondrial ATP synthase oligomers to safeguard mitochondrial function

Rubén Quintana-Cabrera[1,2,9,10,11], Charlotte Quirin[1,2], Christina Glytsou[1,2,12], Mauro Corrado[1,2,13], Andrea Urbani[3], Anna Pellattiero[1,2], Enrique Calvo[4], Jesús Vázquez [4], José Antonio Enríquez [4,5], Christoph Gerle[6,7], María Eugenia Soriano [8], Paolo Bernardi [3,8] & Luca Scorrano [1,2]

It is unclear how the mitochondrial fusion protein Optic atrophy 1 (OPA1), which inhibits cristae remodeling, protects from mitochondrial dysfunction. Here we identify the mitochondrial $F_1F_o$-ATP synthase as the effector of OPA1 in mitochondrial protection. In OPA1 overexpressing cells, the loss of proton electrochemical gradient caused by respiratory chain complex III inhibition is blunted and this protection is abolished by the ATP synthase inhibitor oligomycin. Mechanistically, OPA1 and ATP synthase can interact, but recombinant OPA1 fails to promote oligomerization of purified ATP synthase reconstituted in liposomes, suggesting that OPA1 favors ATP synthase oligomerization and reversal activity by modulating cristae shape. When ATP synthase oligomers are genetically destabilized by silencing the key dimerization subunit $e$, OPA1 is no longer able to preserve mitochondrial function and cell viability upon complex III inhibition. Thus, OPA1 protects mitochondria from respiratory chain inhibition by stabilizing cristae shape and favoring ATP synthase oligomerization.

[1] Venetian Institute of Molecular Medicine, 35129 Padua, Italy. [2] Department of Biology, University of Padua, 35121 Padua, Italy. [3] Department of Biomedical Sciences, University of Padua, Padua 35121, Italy. [4] Centro Nacional de Investigaciones Cardiovasculares Carlos III, 28029 Madrid, Spain. [5] CIBERFES, Institute of Health Carlos III, Madrid, Spain. [6] Institute for Protein Research, Osaka University, Suita, Osaka, Japan. [7] Core Research for Evolutional Science and Technology, Japan Science and Technology Agency, Kawaguchi, Japan. [8] Institute of Neuroscience, Consiglio Nazionale delle Ricerche, Padua, Italy. [9]Present address: University of Salamanca, Consejo Superior de Investigaciones Científicas CSIC, Institute of Functional Biology and Genomics, Salamanca 37007, Spain. [10]Present address: Institute of Biomedical Research of Salamanca, University Hospital of Salamanca, University of Salamanca, CSIC, 37007 Salamanca, Spain. [11]Present address: CIBERFES, Institute of Health Carlos III, Madrid, Spain. [12]Present address: Department of Pathology, NYU School of Medicine, 10016 New York, NY, USA. [13]Present address: Max Planck Institute of Immunology and Epigenetics, 79108 Freiburg im Breisgau, Germany. Correspondence and requests for materials should be addressed to L.S. (email: luca.scorrano@unipd.it)

Cristae are a pleomorphic subcompartment of the inner mitochondrial membrane (IMM) essential for biological energy conversion and for regulation of mitochondrial apoptosis[1]. Cristae shape and mitochondrial function are intimately connected[2]: changes in cristae morphology affect stability of respiratory supercomplexes (RCS)[3,4], functional quaternary assemblies of respiratory chain complexes (RCC)[1,5–8]. In a reductionist view, cristae shape can be defined by the curvature of two regions: the cristae junctions (CJs), narrow tubular structures connecting cristae to the inner boundary membrane[9]; and the cristae lumen proper. CJs limit the diffusion of cytochrome c from the cristae[10,11] and their enlargement (i.e., the transition from the negative curvature of a shape resembling a solid hyperboloid to the null curvature of a cylinder or to the positive curvature of a spheroid) triggered by proapoptotic BH3-only BCL-2 family members BID, BIM-S, or BNIP3 allows cytochrome c redistribution[10,12–14]. During this process of cristae remodeling, cristae lumen width is also altered: the transition to a more positive CJ curvature results in overall cristae widening. Functionally, cristae widening destabilizes RCS and reduces mitochondrial oxidative phosphorylation efficiency[3]. In sum, cristae shape is a key morphological parameter that influences mitochondrial apoptosis and respiration.

A central modulator of cristae curvature is the IMM dynamin-related protein Optic atrophy 1 (OPA1): OPA1 oligomers maintain a negative CJ curvature, controlling cytochrome c redistribution and release[11,13] and stabilizing RCS to increase respiratory efficiency[3,4,15]. Because of these pleiotropic effects on mitochondrial function, controlled OPA1 overexpression is beneficial against a variety of pathological conditions, ranging from ischemia-reperfusion to massive hepatocellular apoptosis to muscular atrophy[4], and even to deletion of genes essential for assembly of RCC[15]. This remarkable protective effect suggests that OPA1 does not work alone in the regulation of mitochondrial structure and hence function.

Other key players in cristae morphology include the MICOS complex, a multiprotein structure conserved from yeast to mammals[16] and the mitochondrial $F_1F_o$-ATP synthase, whose oligomers are retrieved on the edges of the cristae and contribute to define cristae curvature[17–25]. In mammals, OPA1 and the core MICOS component Mic60 physically interact and cooperate to stabilize the negative CJs curvature; however, CJ and cristae diameter, the key parameters defining mitochondrial apoptosis and respiration, are solely controlled by OPA1[26], ruling out a role for MICOS in the mitochondrial protection afforded by OPA1. Another potential candidate is the mitochondrial ATP synthase that also can physically interact with OPA1[27,28]. The ATP synthase utilizes the mitochondrial proton electrochemical gradient ($\Delta\mu_{H+}$) generated by the RCC[1,29,30] to recycle ATP from ADP and Pi[31,32]. This enzyme also forms the permeability transition pore (PTP)[33], a large conductance channel whose prolonged openings collapse $\Delta\mu_{H+}$ and cause cell death. Finally, the ATP synthase can also sustain the $\Delta\mu_{H+}$ when RCC are inhibited, by running in its reverse mode, hydrolyzing ATP to pump protons across the IMM[32].

However, the interplay between ATP synthase and OPA1 in mitochondrial ultrastructure and function is unclear. Here, we provide evidence that OPA1 requires ATP synthase oligomers to protect mitochondria from respiratory chain inhibition.

## Results

### OPA1 counteracts mitochondrial dysfunction by antimycin A.
OPA1 overexpression is beneficial in vivo against primary and secondary mitochondrial dysfunction[4,15] because of its ability to blunt mitochondrial apoptosis[11] and to promote RCS stability[3].

However, the extent and mechanism of this mitochondrial protection are unclear. We therefore capitalized on models of Opa1 mild overexpression and conditional ablation to investigate in real time how OPA1 levels influenced the mitochondrial electrochemical gradient in response to complex III blockage by the inhibitor antimycin A (AA)[34,35].

In mouse adult fibroblasts (MAFs) from Opa1 transgenic (Opa1$^{tg}$) mice, OPA1 levels and oligomers were 1.5-fold higher (Supplementary Fig. 1a, b) and cristae accordingly ~30% narrower (Supplementary Fig. 1c, d), as previously reported[3]. Real-time measurements of the fluorescence of the potentiometric dye tetramethylrhodamine methylester (TMRM) indicated that Opa1$^{tg}$ MAFs were surprisingly protected by AA-induced mitochondrial depolarization (Fig. 1a, b). We further tested whether OPA1 was able to prevent matrix acidification in the same experimental conditions, by measuring matrix pH with mtSypHer, a ratiometric genetically encoded pH sensor targeted to the mitochondrial matrix[34]. Matrix pH was superimposable in WT and Opa1$^{tg}$ MAFs, irrespective of whether the cells were cultured in media containing glucose or galactose (to force mitochondrial ATP production, Supplementary Fig. 1e). Real-time mtSypHer imaging indicated that OPA1 overexpression blunted also the matrix acidification caused by AA (Fig. 1c, d). We next turned to Opa1$^{flx/flx}$ MAFs where 48 h after transfection with CRE recombinase OPA1 was almost completely absent (Supplementary Fig. 2). In this same timeframe, Opa1 deletion does not modify mitochondrial DNA (mtDNA) content or translation[1,3,36]. Real-time mtSypHer imaging revealed that matrix acidification induced by AA was more severe upon Opa1 deletion (Fig. 1e, f). Thus, OPA1 protects from electrochemical gradient loss upon CIII blockage.

### OPA1 sustains mitochondrial function via ATP synthase activity.
We wondered how a dynamin-related protein involved in cristae morphogenesis and membrane fusion could regulate mitochondrial electrochemical gradient. Because reversal of ATP synthase activity can sustain mitochondrial membrane potential upon RCC inhibition[37], we tested if OPA1 facilitated reversal ATP synthase activity to extrude protons from the matrix and maintain $\Delta\mu_{H+}$. The protective effect of OPA1 overexpression on AA-induced matrix acidification and depolarization was fully abolished by the ATP synthase inhibitor oligomycin (Fig. 2a–d). Oligomycin also equalized AA-induced pH changes in empty vector (EV) and CRE-transfected Opa1$^{flx/flx}$ cells (Fig. 2e, f). The effects of OPA1 on pH could be a consequence of decreased proton leak, or of increased proton pumping by the reversal mode of the ATP synthase. If proton leak were reduced in Opa1$^{tg}$ MAFs, ATP synthase inhibition would result in a higher $\Delta\mu_{H+}$ compared to WT cells. However, membrane potential and matrix pH changes were superimposable in WT and Opa1$^{tg}$ MAFs treated with oligomycin (Supplementary Fig. 3a-c), indicating a similar proton leak in these two cell lines. Thus, the differences in $\Delta\mu_{H+}$ upon CIII inhibition recorded in WT and Opa1$^{tg}$ MAFs could be due to different stimulation of the reversal ATP synthase activity.

To directly measure if OPA1 affected ATP synthase activity in situ, we monitored ATP synthase-dependent hydrolysis of mitochondrial ATP upon complex III inhibition in real time in living cells. The FRET probe ATeam1.03[38] was correctly targeted to the mitochondrial matrix and revealed that basal mitochondrial ATP content was similar in WT and Opa1$^{tg}$ MAFs (Supplementary Fig. 3d). Following complex III inhibition, ATeam1.03 fluorescence decayed 3-fold faster in Opa1$^{tg}$ MAFs, suggesting faster ATP hydrolysis. These fluorescence changes were indeed due to ATP synthase activity, because they were

abolished by the ATP synthase inhibitor oligomycin (Fig. 2g–i and Supplementary Movie 1). Notably, oligomycin did not affect OPA1 oligomerization in WT or *Opa1^tg* MAFs (Supplementary Fig. 4). Thus, OPA1 overexpression stimulates reversal ATP synthase activity.

**OPA1 stabilizes ATP synthase oligomers**. How does OPA1 overexpression impinge on ATP synthase to protect mitochondria from AA? Because OPA1 overexpression favors

mitochondrial RCS assembly by impinging on cristae shape[3], we verified if the same was also true for ATP synthase oligomerization. Blue Native Gel Electrophoresis (BNGE) indicated that oligomeric ATP synthase was stabilized in mitochondria from *Opa1^tg* cells (Fig. 3a, b). When we acutely deleted *Opa1* by expressing Cre recombinase in *Opa1^flx/flx* MAFs, ATP synthase dimers and monomers were less abundant, with an increase in free $F_1$ subunit and a reduction in ATP synthase activity and total protein levels (Fig. 3c–e). Thus, OPA1 levels correlate with ATP synthase oligomerization and stability.

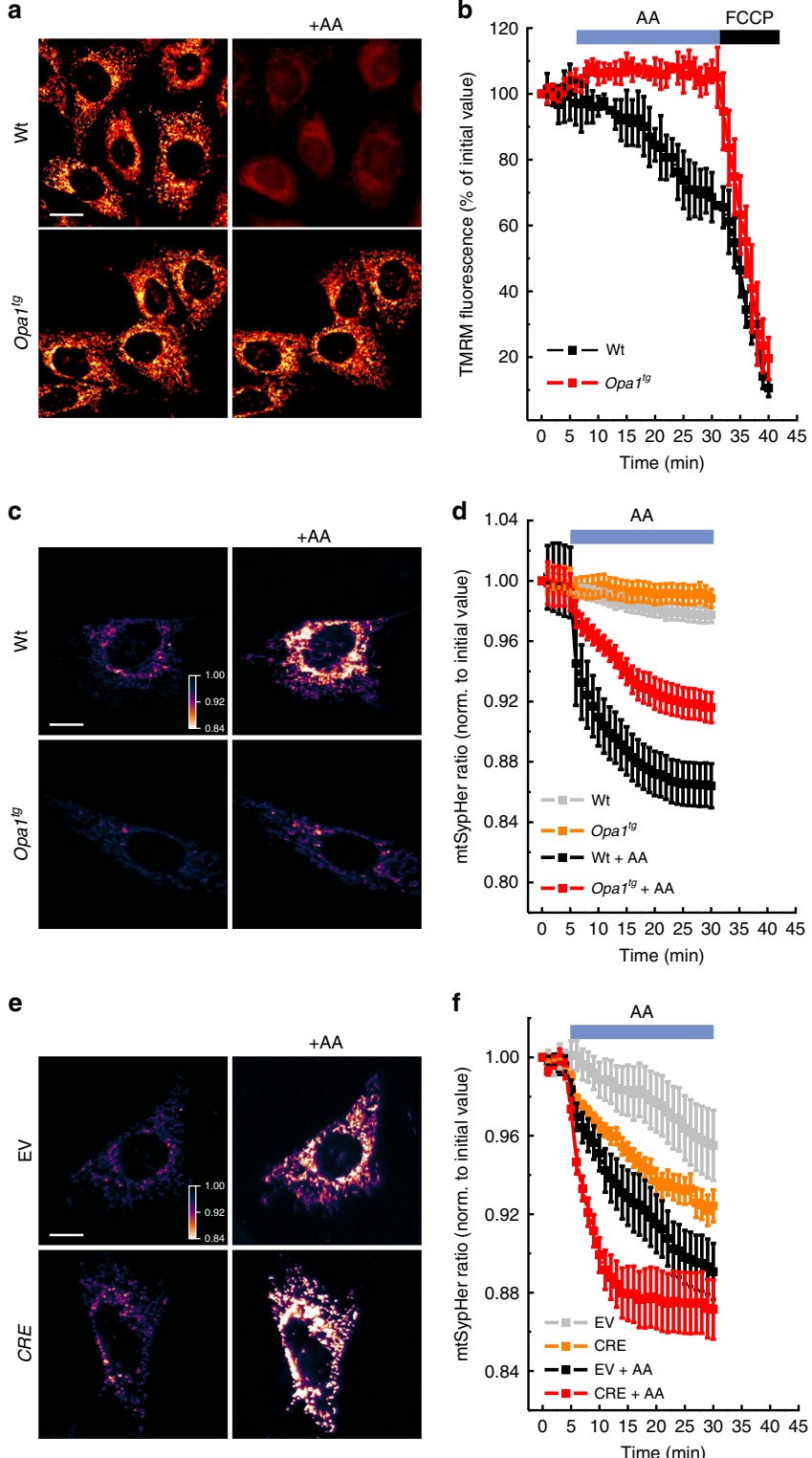

We next verified if OPA1 could stabilize ATP synthase oligomers also during apoptotic cristae shape changes induced by the caspase-8 cleaved, active form of the proapoptotic Bcl-2 family member BID. To this end, we compared the effects on ATP synthase levels and oligomerization of WT BID, of a BID mutant (BID^KKAA) that permeabilizes the mitochondrial outer membrane (OMM) but does not remodel cristae[3] and of an OMM-permeabilization deficient BID mutant (BID^G94E) that causes cristae remodeling but not cytochrome c release[3,39]. While total levels of ATP synthase were not affected by treatment of WT or $Opa1^{tg}$ mitochondria with the different BID mutants (Fig. 3f, g), BID and BID^G94E destabilized ATP synthase oligomers only in WT mitochondria. As expected, treatment with the cristae remodeling deficient BID^KKAA mutant did not affect the ATP synthase oligomerization pattern (Fig. 3f, h).

To further corroborate these biochemical experiments, we turned to an approach of proteomic profiling of high molecular weight (HMW) complexes (complexomic) that allowed us to identify dynamic changes of MICOS during cristae remodeling[26]. We therefore performed a similar analysis focused on ATP synthase complexes. First, we confirmed by immunoblotting and complexomic analysis of native protein complexes separated by BNGE from purified heart mitochondria that treatment with WT BID, but not BID^KKAA, caused the disassembly of OPA1-containing HMW complexes[26] (Fig. 4a–c). We next focused our attention on the oligomeric ATP synthase assemblies. We found that in the BNGE regions corresponding to ATP synthase dimers and oligomers, the spectral counts corresponding to the $F_1$ core subunits ATP5A (α), ATP5B (β), ATP5C1 (γ), ATP5D (δ) and ATP5O (OSCP; see details on the analyzed subunits in supplementary Data 1) were reduced in mitochondria treated with WT BID but not with BID^KKAA (Fig. 4d). The accuracy of this complexomic profiling was further supported by the analysis of the ATP5K (e) subunit essential for ATP synthase dimerization[40–43]: ATP5K was retrieved only in the BNGE region corresponding to dimers and its spectral counts dropped to ~40% of control mitochondria upon BID-treatment; the drop was caused by cristae remodeling, because the spectral counts remained ~72% of the untreated in mitochondria challenged with BID^KKAA (Fig. 4e), as further confirmed by immunoblotting (Fig. 4f). Overall, the analysis of the median of ATP synthase subunits spectral counts confirmed that ATP synthase dimers and oligomers were reduced in BID- but not in BID^KKAA-treated mitochondria (Fig. 4g). In conclusion, independent biochemical approaches indicate that ATP synthase oligomers are stabilized upon OPA1 overexpression and destabilized when $Opa1$ is deleted, or cristae remodeled with concomitant OPA1 HMW oligomers disassembly.

**OPA1 does not directly stimulate ATP synthase oligomerization.** Because OPA1 levels are proportional to ATP synthase

oligomerization, we wished to understand how OPA1 sustained ATP synthase oligomerization. Levels of the ATP synthase inhibitory factor (IF₁), whose oligomerization is influenced by matrix pH[44] and that in its oligomeric form can stabilize ATP synthase dimers[45], were similar in WT and $Opa1^{tg}$ MAFs (Supplementary Fig. 5), suggesting that other mechanisms were in place.

We first asked whether OPA1 could directly interact with ATP synthase. Chemical crosslinking using 1-Ethyl-3-(3-dimethylaminopropyl)carbodiimide hydrochloride (EDC) resulted in the appearance of a ~130 KDa band immunoreactive for both ATP5B (β-subunit) and OPA1 (Fig. 5a). Moreover, ATP5B and OPA1 reciprocally co-immunoprecipitated (Fig. 5b, c), confirming and extending previous results that identified several ATP synthase subunits as OPA1 interactors[28]. To verify if OPA1 could also directly promote or stabilize the oligomeric forms of ATP synthase, we turned to an in vitro system of purified proteins and liposomes. We prepared proteoliposomes containing highly pure ATP synthase stabilized in the high affinity lipid like detergent Lauryl Maltose Neopentyl Glycol, as confirmed by EM (Fig. 5d). We also established a protocol to produce and purify a recombinant form of soluble OPA1 (lacking the import sequence and the transmembrane domain, supplementary Figure 6a) by adding a 6-His-Tag to its C-terminus. Following induction of expression in competent bacteria we purified recombinant soluble OPA1 (rOPA1) by affinity chromatography on Ni-NTA beads. Since a contaminant protein with ATPase activity bound to rOPA1, we removed it by extensive washes with ATP, before we eluted by increasing imidazole concentrations obtaining moderate yields of rOPA1 devoid of the contaminating ATPase activity (Supplementary Fig. 6b). Enzymatic activity of the dialyzed rOPA1 determined by reverse phase chromatography was in the range of other dynamin-related GTPases, confirming that rOPA1 was active and stable. We therefore incorporated rOPA1 into the proteoliposome lumen, mimicking the relative topology of OPA1 and ATP synthase in mitochondria (Supplementary Fig. 6c). When we analyzed these proteoliposomes by BNGE, we did not observe any effect of rOPA1 on ATP synthase oligomerization (Fig. 5e, f). In a further in vitro experiment, we added increasing concentrations of rOPA1 to native purified ATP synthase in a digitonin based buffer. However, also in these conditions rOPA1 failed to stimulate ATP synthase dimerization; if anything, in the presence of rOPA1, ATP synthase was mostly monomeric (Fig. 5g, h). Thus, despite its interaction with ATP synthase subunits, OPA1 does not directly stimulate or stabilize ATP synthase oligomerization.

**OPA1 sustains mitochondrial function via ATPase oligomers.** In $Opa1^{tg}$ mitochondria ATP synthase oligomers are more abundant and matrix pH is maintained upon complex III inhibition. We therefore turned to a genetic approach to test whether OPA1 required ATP synthase oligomers to protect from

---

**Fig. 1** OPA1 prevents mitochondrial electrochemical gradient loss caused by CIII inhibition. **a** Representative color-coded frames from real time imaging of TMRM fluorescence in MAFs of the indicated genotype. Where indicated, cells were treated for 30 min with 10 μM antimycin A (AA). Scale bar, 20 μm. **b** Quantitative analysis of TMRM fluorescence over mitochondrial regions in real time imaging experiments as in **a**. Where indicated, cells were treated with AA (10 μM) and with FCCP (2 μM). Data are average ± SEM. ($n = 5$ for each group). **c** Representative color-coded frames from real time imaging of mtSypHer fluorescence ratio in MAFs of the indicated genotype. Where indicated, cells were treated for 30 min with 10 μM AA. Rainbow color bar: pseudocolor scale of mtSypHer fluorescence ratio. Scale bar, 20 μm. **d** Quantitative analysis of mitochondrial mtSypHer fluorescence ratio in real time imaging experiments as in **c**. Where indicated, cells were treated with AA (10 μM). Data are mean ± SEM of at least four independent experiments. **e** Representative color-coded images of mitochondrial mtSypHer fluorescence in $Opa1^{flx/flx}$ cells transfected with empty (EV) or CRE-encoding vectors and treated where indicated for 30 min with AA (10 μM). Rainbow color bar: pseudocolor scale of SypHer fluorescence ratio. Scale bar, 20 μm. **f** Quantitative analysis of mitochondrial mtSypHer fluorescence ratio in real time imaging experiments as in **a**. Data are mean ± SEM of at least four different experiments

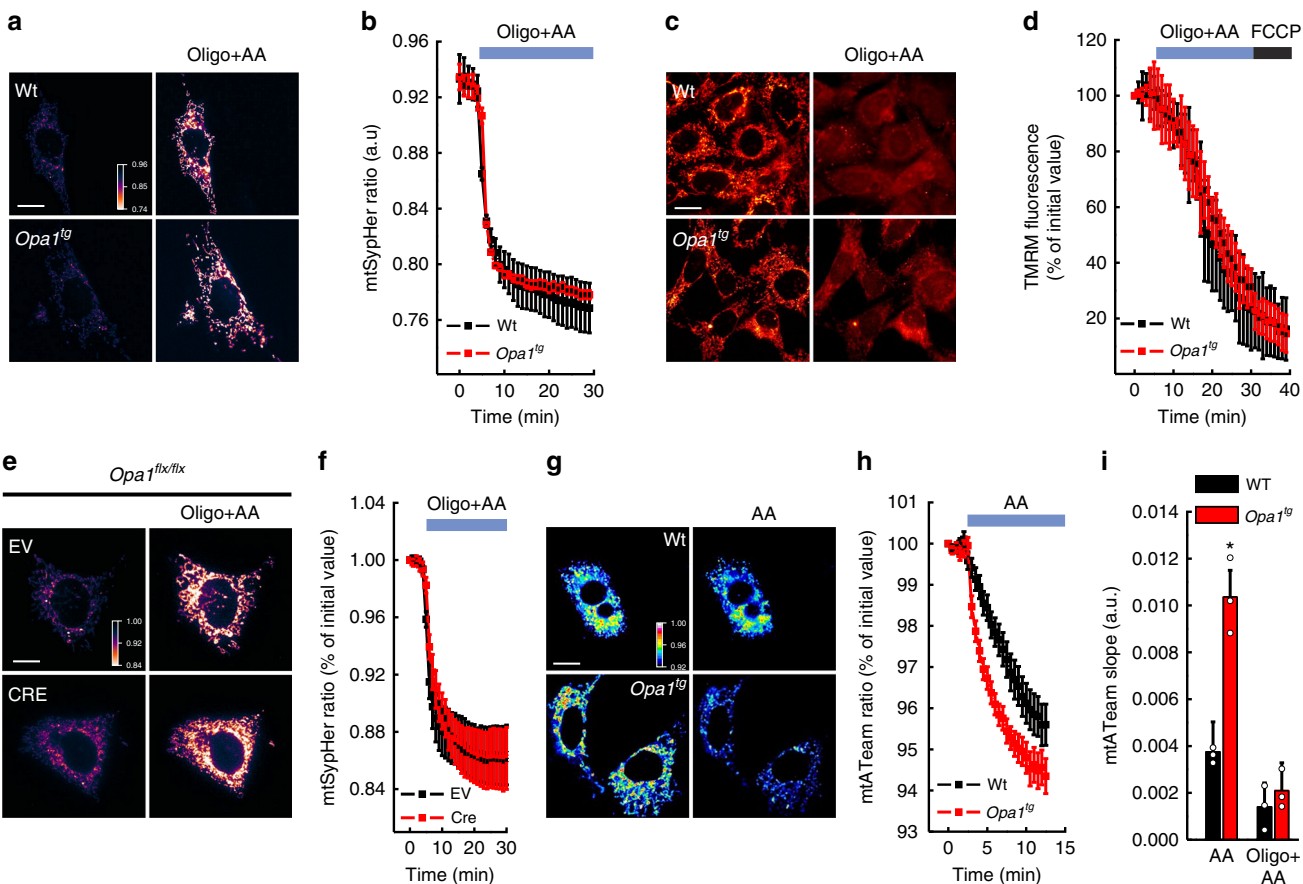

**Fig. 2** OPA1 requires ATP synthase activity to prevent mitochondrial ΔpH loss caused by CIII inhibition. **a** Representative color-coded frames from real time imaging of mtSypHer ratio fluorescence in MAFs of the indicated genotype. Where indicated, cells were incubated with oligomycin (oligo, 1 μM for 30 min) and antimycin A (AA) (10 μM added 5 min after oligo, for 25 min). Rainbow color bar: pseudocolor scale of SypHer fluorescence ratio. Scale bar, 20 μm. **b** Quantitative analysis of mitochondrial mtSypHer fluorescence ratio in real time imaging experiments as in **a**. Data are mean ± SEM of at least four independent experiments. **c** Representative pseudocolor-coded frames from real time imaging of TMRM fluorescence in MAFs of the indicated genotype. Where indicated, cells were incubated with oligomycin (1 μM for 5 min, before AA additions) and AA (2 μM for 25 min). Scale bar, 20 μm. **d** Quantitative analysis of mitochondrial TMRM fluorescence ratio in real time imaging experiments as in **c**. Where indicated, cells were treated with 2 μM carbonyl cyanide 4-(trifluoromethoxy)phenylhydrazone (FCCP). Data are mean ± SEM of at least four independent experiments. **e** Representative color-coded images of mitochondrial mtSypHer fluorescence in Opa1$^{flx/flx}$ cells transfected with empty (EV) or Cre-encoding vectors and treated for 30 min with AA (10 μM) and oligomycin (1 μM, added 5 min before AA) where indicated. Rainbow color bar: pseudocolor scale of mtSypHer fluorescence ratio. Scale bar, 20 μm. **f** Quantitative analysis of mitochondrial mtSypHer fluorescence ratio in real time imaging experiments as in **e**. Data are mean ± SEM of at least 4 independent experiments. **g** Pseudocolor-coded frames of mtATeam FRET fluorescence ratio in cells of the indicated genotype. Where indicated, cells were treated for 10 min with AA (10 μM). Rainbow color bar: pseudocolor scale of ATeam fluorescence ratio. Scale bar, 20 μm. **h** Quantitative analysis of mitochondrial mtATeam fluorescence ratio in real time imaging experiments as in **g**. Data are mean ± SEM of at least three independent experiments. **i** Mean ± SEM of mtATeam fluorescence ratio slopes, calculated from experiments as in **g**. Where indicated, oligomycin (1 μM) was added 5 min before AA. *$p < 0.05$ in an unpaired two-sample Student's $t$ test

mitochondrial CIII blockage. ATP synthase subunit $e$ (ATP5k) is required for ATP synthase dimerization[40–43], but its ablation does not compromise monomer function[40,46,47]. Efficient *ATP5k* silencing in WT and *Opa1$^{tg}$* MAFs (Fig. 6a) yielded a super-imposable reduction in ATP synthase dimers/total ATP synthase ratio (Fig. 6b, c)[41]. Because subunit $e$ downregulation causes mitochondrial ultrastructural defects[19,24,48], we inspected cristae shape in *ATP5k* silenced mitochondria. We morphometrically evaluated three parameters: cristae width, cristae number, and the ratio between CJ and cristae number, an indicator of the presence of arched or septate cristae, i.e., cristae connecting the IMM at two or more CJ. While *ATP5k* downregulation did not affect cristae width (Fig. 6d, e), it reduced cristae number and it increased the CJ/cristae number ratio (Fig. 6d, f, g), indicating the formation of arched and septate cristae with less tips[41]. In *Opa1$^{tg}$*

MAFs the formation of these arched/septate cristae was blunted (Fig. 6g). Interestingly, *ATP5k* silencing reduced OPA1 oligomers in WT but not in *Opa1$^{tg}$* MAFs (Supplementary Fig. 7a, b), suggesting that the formation of arched cristae caused by *ATP5k* silencing in WT cells depends on disassembly of OPA1 oligomers that are more stable upon OPA1 overexpression[4,11]. Our results indicate that OPA1 can compensate for the ultrastructural defects caused by reduced ATP synthase dimerization. A diametrically different scenario emerged when we delved into the mechanism of *Opa1$^{tg}$* cells protection from AA-induced mitochondrial depolarization. *ATP5k* downregulation did not reduce resting ATP content in WT or *Opa1$^{tg}$* cells (Supplementary Fig. 8a)[40,47]. Conversely, it completely abolished the effect of OPA1 on protection from AA-induced depolarization (Fig. 7a–c and Supplementary Movie 2), as well as on stimulation of ATPase reversal

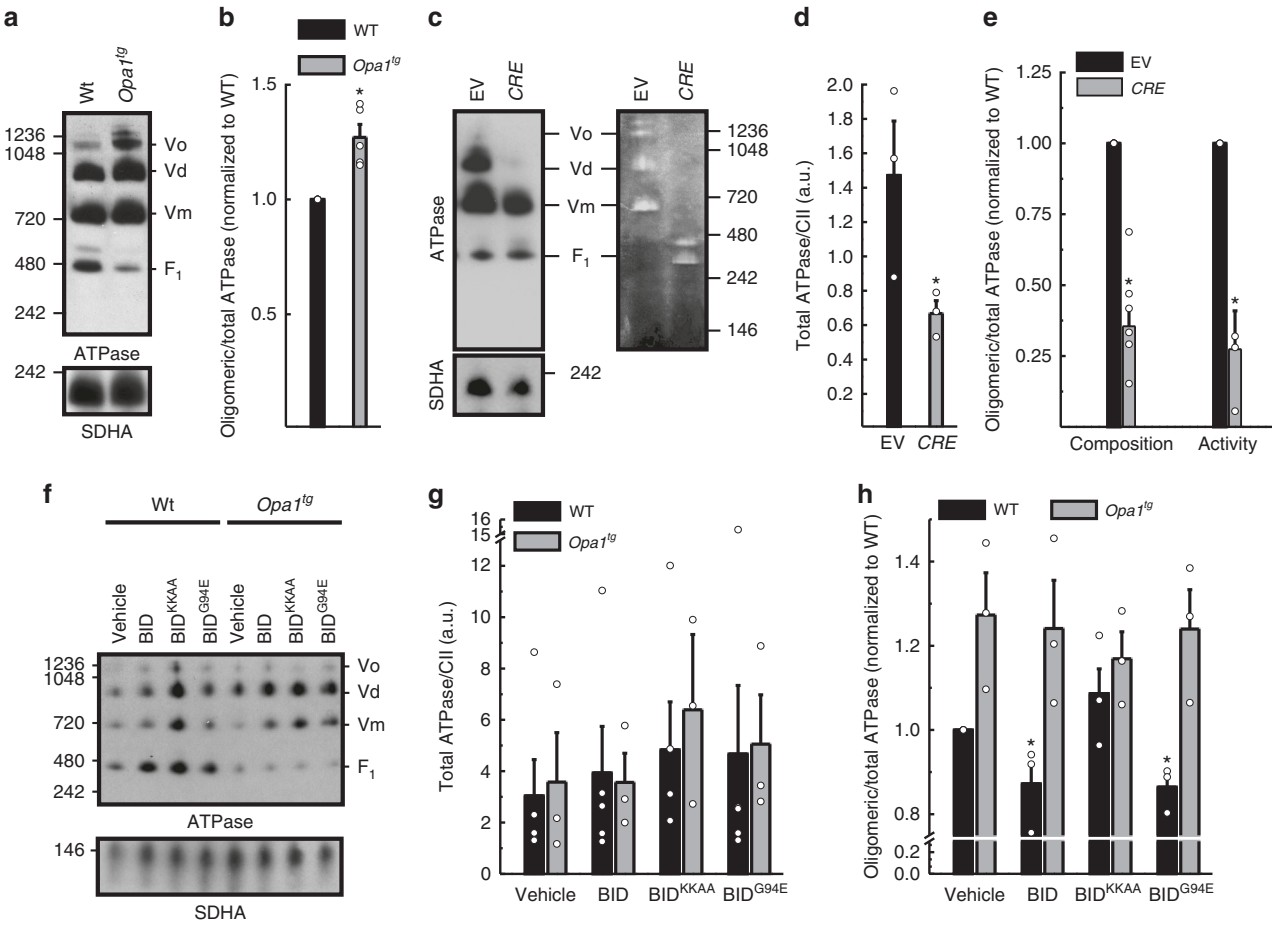

**Fig. 3** OPA1 promotes stabilization of ATP synthase oligomers. **a** Equal amounts (40 µg) of digitonin-solubilized mitochondrial extracts from MAFs of the indicated genotype were separated by BNGE and immunoblotted with anti-ATP synthase subunit α (ATPase) and Succinate Dehydrogenase (SDHA) antibodies. ATPase oligomers (Vo), dimers (Vd), monomers (Vm) and $F_1$ are indicated. **b** Densitometric analysis of experiments performed as in **a**. Data represent mean ± SEM of Vo + Vd (oligomeric)/total ATP synthase (Vo + Vd + Vm + $F_1$) from five different experiments. *$p < 0.05$ in a two-way ANOVA vs. control. **c** Equal amounts (40 µg) of digitonin-solubilized mitochondrial extracts from MAFs of the indicated genotype were separated by BNGE and immunoblotted with the indicated antibodies (left) or processed for ATPase activity (right). ATPase oligomers (Vo), dimers (Vd), monomers (Vm) and $F_1$ are indicated. **d**, **e** Quantitative densitometric analysis of total ATP synthase/CII (SDHA) (**d**) and of oligomeric/total ATP synthase conformations (**e**) in experiments as in **c**. Data show mean ± SEM of at least three independent experiments. *$p < 0.05$ in a two-way ANOVA versus EV. **f** Mitochondria from MAFs of the indicated genotypes were treated with recombinant BID as indicated for 30 min, lysed and equal amounts (40 µg) of digitonin-solubilized extracts were separated by BNGE and immunoblotted with the indicated antibodies. **g**, **h** Quantitative densitometric analysis of total/CII (SDHA) (**g**) and of oligomeric/total ATP synthase conformations (**h**) in experiments as in **f**. Data are normalized to untreated cells and represent mean ± SEM of at least three independent experiments. *$p < 0.05$ in an unpaired two-sample Student's $t$ test versus untreated

function, measured by following matrix ATP hydrolysis in real time (Fig. 7d, e; Supplementary Fig. 8b). Thus, OPA1 requires ATP5k and hence efficient ATP synthase dimerization to protect from mitochondrial dysfunction. If this model was correct, OPA1 should inhibit cell death induced by complex III inhibition and ATP5k should be essential for this cytoprotective effect. Indeed, WT MAFs grown in galactose-containing media rapidly died when challenged with AA, whereas $Opa1^{tg}$ MAFs were protected; downregulation of *ATP5k* equalized death levels between $Opa1^{tg}$ and WT cells (Fig. 7f). Our results indicate that the protection provided by OPA1 overexpression requires ATP5K and efficient ATP synthase oligomerization to sustain mitochondrial ΔpH and curtail cell death following complex III Inhibition.

## Discussion

How the IMM fusion and cristae biogenesis protein OPA1 preserve mitochondrial function from a plethora of tissue-damaging insults and from respiratory chain inhibition is unclear. Here, multiple lines of evidence point to a key role for ATP synthase oligomerization status and reversal (i.e., ATP hydrolase) activity.

OPA1 overexpression counteracts multiple insults including ischemia, atrophy and death-receptor mediated apoptosis that converge on mitochondria[4,49], and most remarkably it corrects electron transport chain defects in vivo[15]. Part of its protective action can be attributed to the inhibition of the apoptotic cristae remodeling pathway, resulting in reduced cytochrome c release and apoptosis[10,12–14], but how OPA1 improves mitochondrial function when RCC are genetically impaired is less clear. Negative cristae, i.e., concave curvature at cristae junctions, correlate with increased RCS stability and increased residual RCC function[15]. While this could explain the correction of mitochondrial function in mitochondria lacking the complex IV assembly factor *Cox15*, it does not explain the improvement observed in mice where the essential complex I subunit *Ndufs4* is deleted[15]. Our results indicate that cristae shaping by OPA1 fosters ATP synthase oligomers and reversal activity, providing a unifying mechanism for

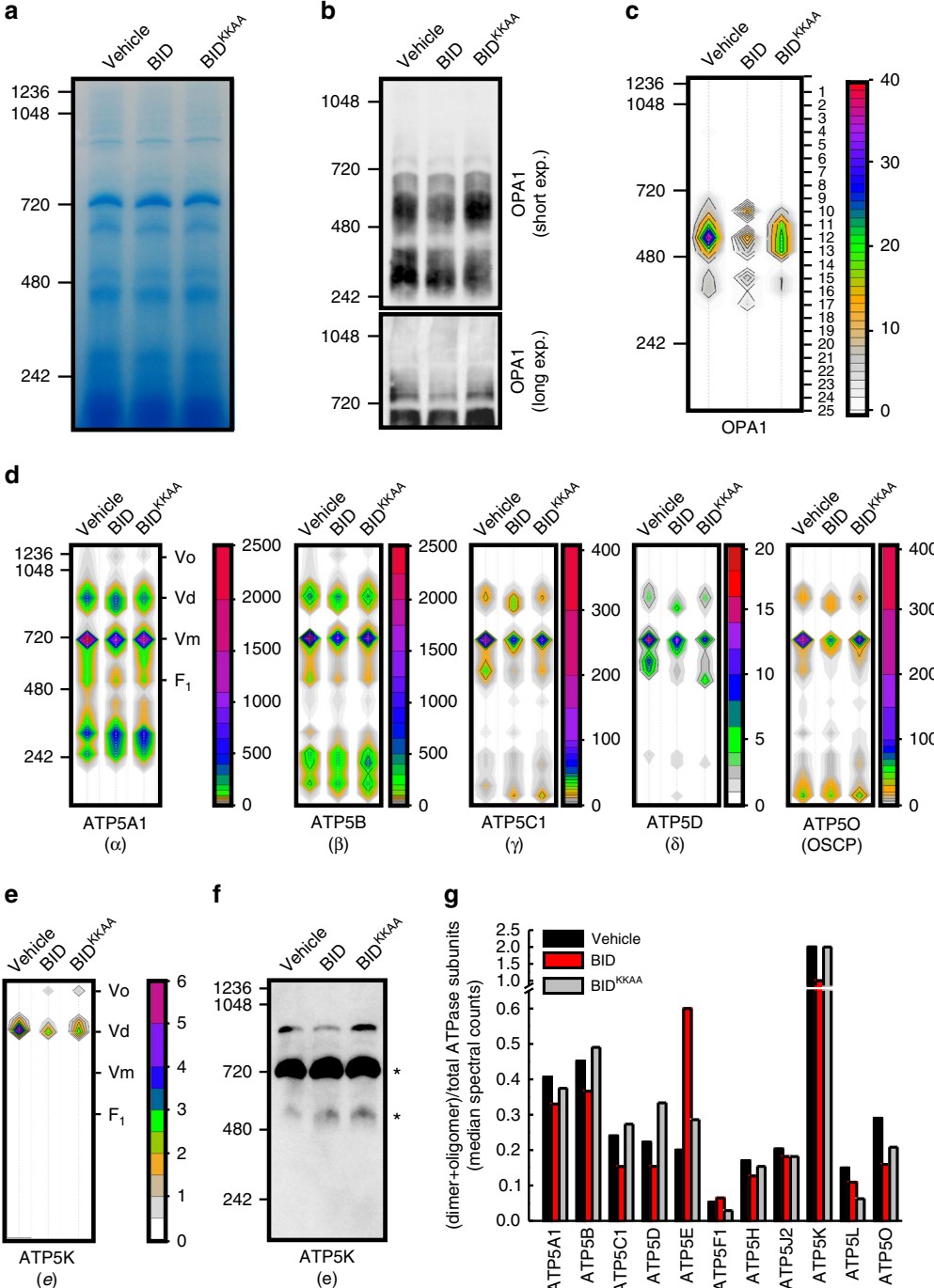

**Fig. 4** ATP synthase oligomers are reduced by cristae remodeling. **a**, **b** Mouse heart mitochondria were incubated as indicated and after 30 min protein lysates (30 μg) were separated by BNGE and Coomassie stained (**a**) or immunoblotted with the indicated antibody (**b**). **c** Representative color-contour plots of spectral counts of OPA1 peptides from quantitative DiS-MS analysis of extracted mitochondrial complexes. Experiments were as in **a**. Numbers indicate the bands excised for MS analysis. The rainbow color scale indicates the number of spectral counts. **d**, **e** Color-contour plot of spectral counts of $F_1$ core components (α, β, γ, δ, OSCP, panel d) and of the $F_O$ ATPase dimerization subunit e (ATP5k) from experiments as in **c**. **f** Experiments were as in **b**, except that membranes were decorated with the anti-ATPase subunit e antibody. Asterisks: cross-reactive unspecific bands. **g** Quantification of experiments as in **d**, **e**. The median values of the spectral counts of the indicated ATP synthase subunits were calculated in mitochondria pooled from three experiments performed as in **a**. The ratio of the median values in the dimer+oligomer region over the total forms is plotted

the ability of OPA1 to sustain mitochondrial electrochemical gradient and function when respiratory chain is inhibited.

Dynamic complexomic analysis of mitochondria undergoing cristae remodeling and OMM permeabilization is a powerful tool to inspect HMW complexes reorganization during cristae shape changes. This method can also identify unexpected OPA1 partners

in cristae morphogenesis, such as the MICOS components Mic60 and Mic19[26] and the SLC25A solute carriers family members that relay respiratory substrates availability to OPA1 to trigger the orthodox to condensed cristae transition[28]. By comparative complexomics we discovered that ATP synthase oligomers are very dynamic and that are affected by cristae shape changes.

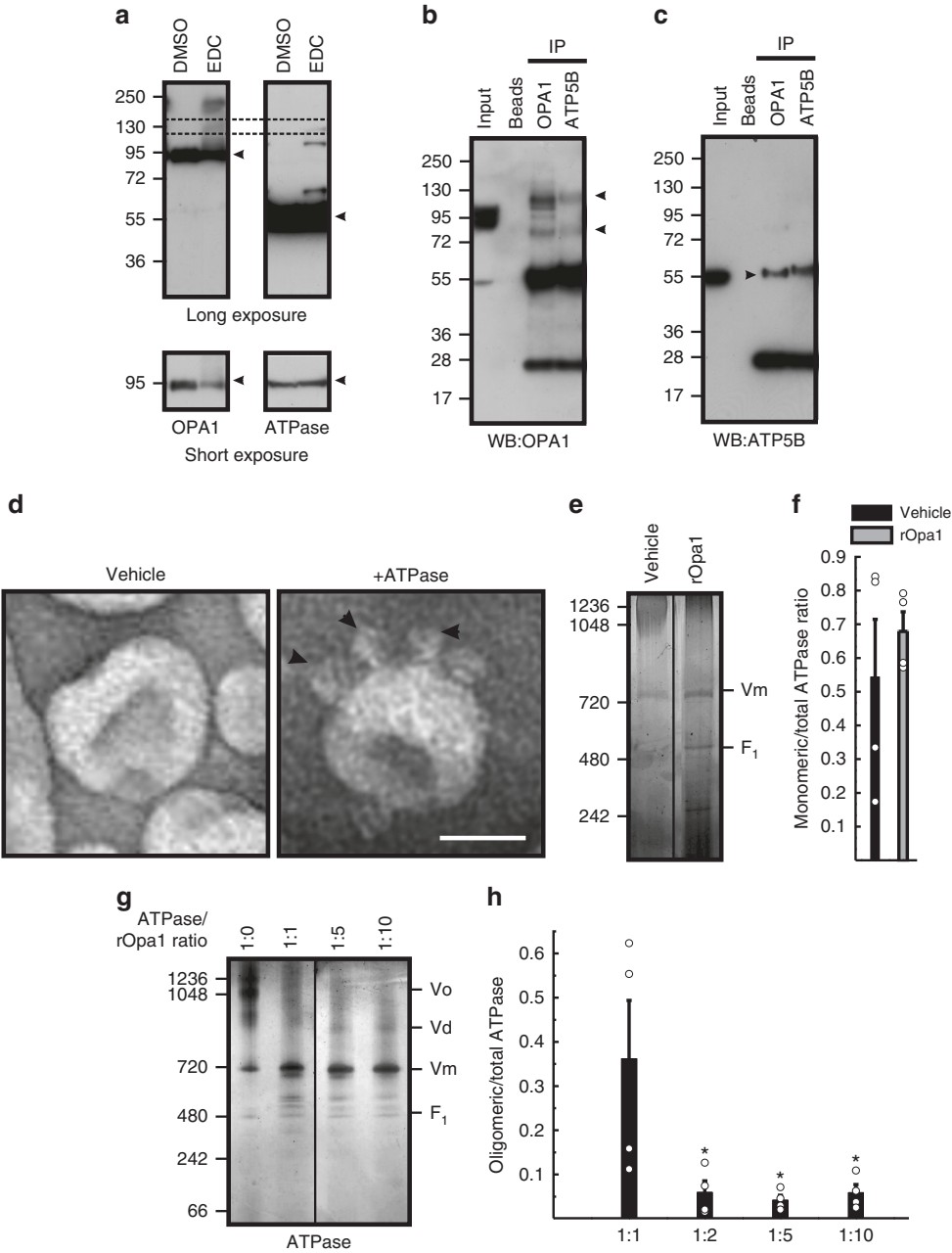

**Fig. 5** OPA1 does not directly stimulate ATP synthase oligomerization. **a** Purified mitochondria were treated where indicated with 10 mM EDC. After 30 min the reaction was quenched, mitochondria lysed and equal amounts (40 μg) of proteins were separated by SDS-PAGE and immunoblotted using the indicated antibodies. Dashed box: OPA1 and ATPase cross-reactive adduct. Arrowheads: non-crosslinked OPA1 and ATPase. Bottom panels: short exposure blots of non-crosslinked OPA1 and ATPase. **b**, **c** Mitochondrial lysates (200 μg) were immunoprecipitated with the indicated antibodies coupled to Protein A agarose beads; bound proteins were separated by SDS-PAGE and immunoblotted using the indicated antibodies. Input was diluted 1:10. **d** Representative EM image of a negatively stained proteoliposome containing rOPA1 in the lumen either in the absence (left panel) or presence (right panel) of recombinant ATP synthase in the membrane. Arrowheads: $F_1$ oriented towards the outside of the liposome. Scale bar, 50 nm. **e** Proteoliposomes harboring ATPase and either empty or containing rOpa1 were solubilized and proteins separated by BN-PAGE. **f** Densitometric quantification of ATPase composition in experiments as in **e**. Data are mean ± SEM of four independent experiments. **g**, **h** Representative BNGE (**g**) and densitometric quantification (**h**) of ATPase oligomeric forms upon incubation of the indicated ratios of rOPA1 with ATP synthase (3 μg). Data are mean ± SEM of four independent experiments. Vo: ATPase oligomers, Vd: dimers, Vm: monomers

A proteomic profiling of OPA1 interactors identified different subunits of ATP synthase[28], suggesting a possible mechanism for the stabilization of the latter. Indeed, OPA1 co-immunoprecipitates with ATP synthase and both are retrieved in high-order crosslinked complexes, where OPA1 could directly stabilize ATP synthase. However, a reductionist approach of purified recombinant OPA1 and ATP synthase containing proteoliposomes failed to prove that OPA1 directly stimulates or stabilizes ATP synthase oligomerization. These experiments suggest that the cristae curvature, promoted by OPA1, might itself stabilize ATP synthase oligomers. This effect would complement that of ATP synthase dimers on membrane curvature, in

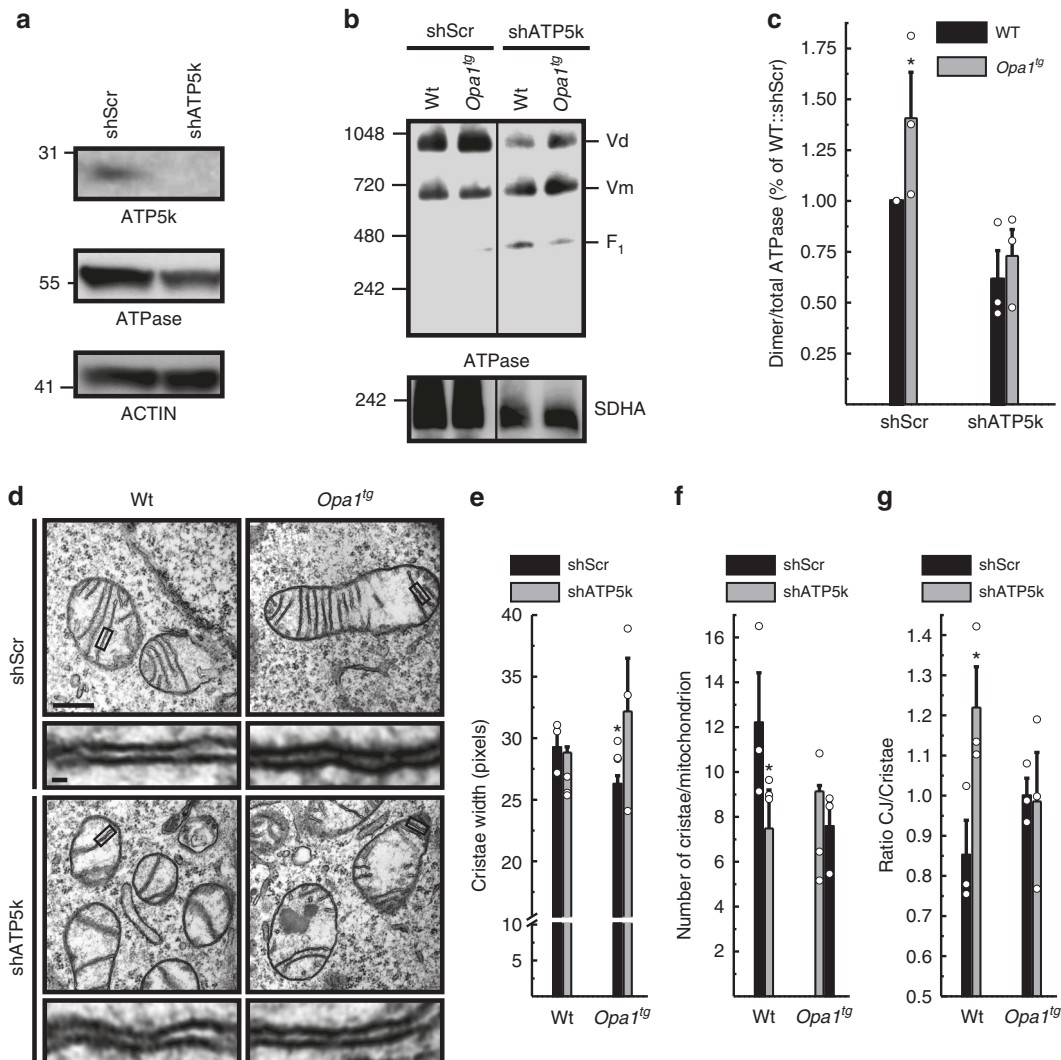

**Fig. 6** OPA1 requires ATP5k to stabilize ATP synthase oligomers. **a** Equal amounts (30 µg) of lysates from MAFs of the indicated genotype transfected for 48 h with the indicated shRNA were separated by SDS-PAGE and immunoblotted with the indicated antibodies (ATPase is ATP5A). **b** Equal amounts (40 µg) of digitonin-solubilized mitochondrial extracts from MAFs of the indicated genotype transfected with the indicated shRNAs were separated by BNGE and immunoblotted with the indicated antibodies. **c** Quantitative densitometric analysis of ATPase dimer vs. total ATPase. Data are mean ± SEM of three independent experiments performed as in **b**. **d** Representative EM micrographs of cells of the indicated genotype transfected with the indicated shRNA and GFP and after 48 h sorted for GFP⁺ and processed for EM. Boxed areas are magnified 12× in the bottom images. Scale bars: 500 nm; 20 nm in bottom magnifications. **e**–**g** Morphometric analysis of cristae maximal width (**e**), number of cristae per mitochondrion (**f**) and cristae junctions per cristae (**g**) in experiments as in **d**. At least 30 mitochondria per experimental condition were analyzed. Mean values ± SEM of three independent experiments are shown. *$p < 0.05$ in a two-way ANOVA versus control (**e**) or paired two-sample Student's $t$ test versus scramble shRNA (shScr) (**f**, **g**)

a feed forward loop that ultimately leads to stabilization of cristae shape by its multiple molecular determinants.

Traditionally, ATP synthase dimerization has been regarded as a cornerstone in cristae morphogenesis: rows of ATP synthase dimers at the edge of the cristae maintain the orthodox structure of the latter[48,50]. In yeast, genetic dimerization disruption results in aberrant mitochondrial ultrastructure[19,23,48]. This organization has also functional consequences: the ATP synthase dimers on the cristae edge contribute to create a proton gradient surrounding the enzyme[24,51], ultimately favoring ATP synthesis[50]. By using apoptotic and genetic manipulations, we demonstrate that ATP synthase oligomerization is also affected by cristae curvature: ATP synthase dimers, like supercomplexes, are destabilized during cristae remodeling and when *Opa1* is ablated, and stabilized by OPA1 overexpression[3].

ATP synthase dimers stabilization improves mitochondrial ultrastructure[48,52], raising the question of whether the effects of OPA1 on cristae shape are secondary to the stabilization of ATP synthase dimers. Ablation of *ATP5k* (subunit e), a supernumerary ATP synthase subunit required to bend the IMM and enable dimer formation, but not essential for cell growth, results in loss of cristae tips, formation of cross-sectional septa and occasional onion-like cristae[19,24,40,41,46,53]. OPA1 overexpression reverted these shape changes, indicating that OPA1 can modulate cristae shape independently of ATP synthase dimers that shape cristae edges and tips[23,48,50]. Our genetic analysis places OPA1 upstream also of the ATP synthase in the pathway that controls cristae width, junctions and number[26]. The combination of in vivo and in vitro experiments presented here suggest that ATP synthase dimerization can also be stabilized by cristae shape, possibly by

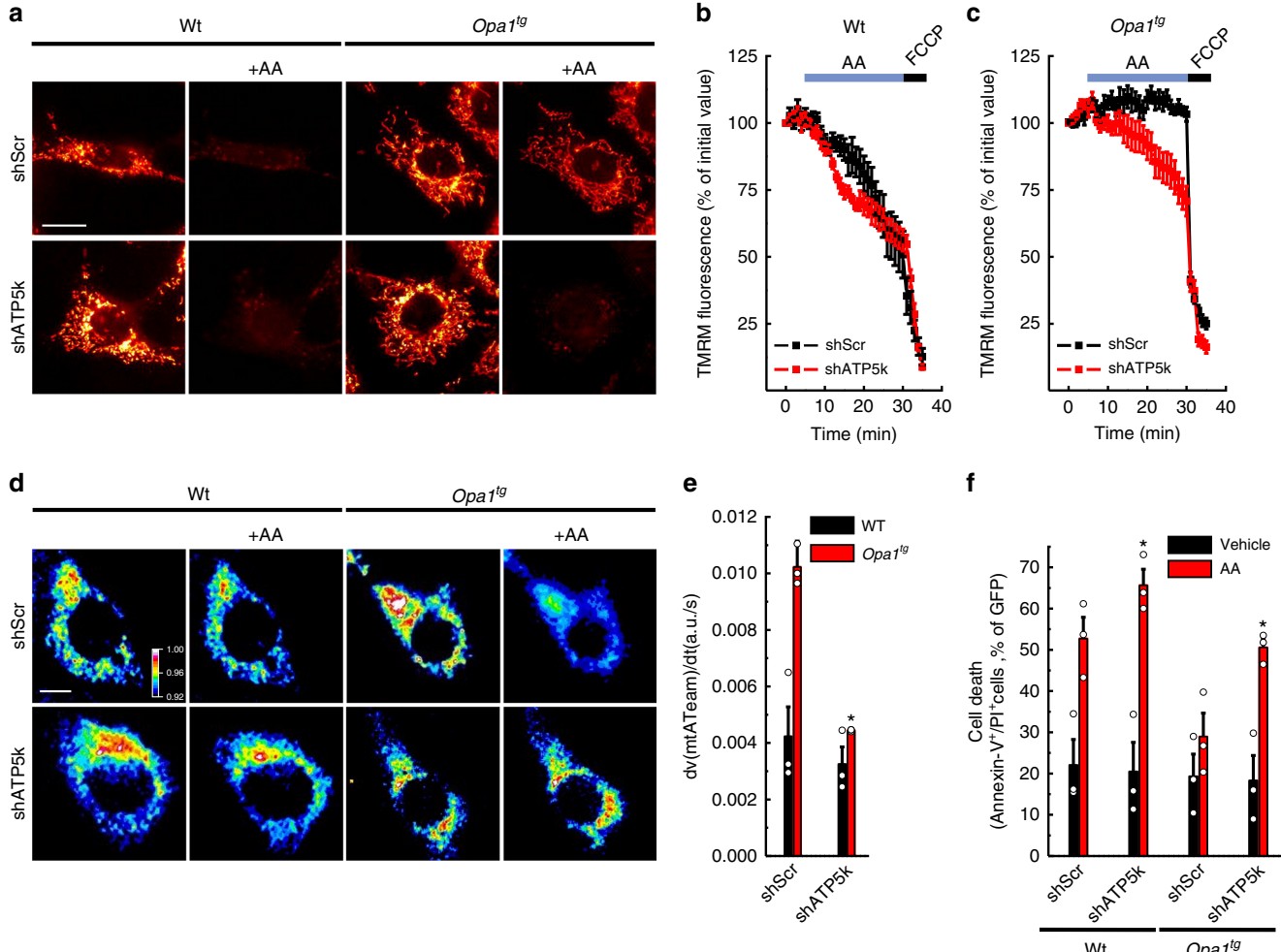

**Fig. 7** OPA1 requires ATP5k to protect from mitochondrial dysfunction. **a** Representative pseudocolor-coded frames from real time imaging of TMRM fluorescence in MAFs of the indicated genotype transfected with the indicated shRNA. Where indicated, cells were incubated with antimycin A (AA) (2 μM for 25 min). Scale bar, 20 μm. **b**, **c** Quantitative analysis of mitochondrial TMRM fluorescence ratio in experiments as in **a**. Where indicated, cells were treated with 2 μM FCCP. Data are mean ± SEM of at least four independent experiments. **d** Pseudocolor-coded frames of mtATeam FRET fluorescence ratio in cells of the indicated genotype. Where indicated, cells were treated for 20 min with AA (10 μM). The rainbow indicates the pseudocolor scale. Scale bar, 20 μm. **e** Mean ± SEM of maximal mtATeam fluorescence ratio slopes in at least three independent experiments as in (**d**). *$p < 0.05$ in an unpaired two-sample Student's $t$ test. versus scramble shRNA (shScr). **f** MAFs of the indicated genotype were cotransfected with GFP and the indicated shRNA and after 72 h were treated with AA (5 μM, 6 h). Data are mean ± SEM of three independent experiments. *$p < 0.05$ in a two-way ANOVA test versus shScr

favoring superassembly in regions of negative curvature generated by OPA1. Conversely, when ATP synthase dimerization was genetically hampered, OPA1 failed to preserve mitochondrial function, indicating a reciprocal functional and structural crosstalk.

*Opa1tg* cells are protected from CIII inhibition in galactose-supplemented media, where the contribution of glycolysis to membrane potential maintenance is marginal[7,54,55]. Mechanistically, OPA1 sustains mitochondrial function by stimulating ATPase reverse activity, which is increased in *Opa1tg* mitochondria: pharmacological inhibition of ATPase activity abolishes the protective function of OPA1. Similarly, genetic inhibition of ATP synthase dimerization abolishes the ATPase activity stimulation and the maintenance of mitochondrial function by OPA1. In this context, it is conceivable that the reduced matrix acidification observed in *Opa1tg* MAFs upon complex III inhibition prevents the binding of the ATPase inhibitor IF$_1$ to the enzyme, which optimally occurs at pH 6.5[44], thereby delaying a IF$_1$-mediated ATPase inhibition and cristae shape stabilization[52,56].

Our work unravels a bioenergetic mechanism accounting for OPA1 protection against mitochondrial failure and suggests that its overexpression might be beneficial to sustain bioenergetics in mitochondriopathies[57–61], germline defects[62] and neurodegeneration[63], whereas OPA1 might be a target to correct oxidative phosphorylation changes in cancer[64–66].

## Methods

**Cell culture and transfection**. WT, *Opa1flx/flx*, *Opa1tg* MAFs were generated in the Scorrano lab and grown in DMEM supplemented with 10% FBS[3]. Unless otherwise stated, glucose in the medium was substituted with 0.9 mg/ml galactose to force ATP production by the respiratory chain[7].

Transient transfection with scramble or ATP5k (F$_1$F$_o$-ATP synthase sub. $e$; NM_007507.2) shRNA encoding SureSilencing® plasmids (Qiagen, KM31364H, plasmid #3) were performed using Transfectin (Biorad). Following overnight transduction, the rate of GFP expression was typically around 60–70%, as determined by flow cytometry. Co-transfections of SureSilencing® shRNA plasmids or pcDNA3.1 vectors harboring WT or mutant tBID cDNAs were performed at a 3:1 ratio to empty pIRES2-eGFP plasmid (Clontech) or vectors encoding for pSypHer-dMito[34] or wt (AT1.03) or inactive (AT1.03R122K/R126K) ATeam[38]. Acute *Opa1* ablation in *Opa1flx/flx* MAFs was performed by co-transfections with the

Cre-recombinase under the control of a PGK promoter (pPGK-Puro, Addgene) and subjected to analysis after 48 h incubations.

**Real time imaging**. For live imaging, cells ($5 \times 10^4$) seeded onto 24-mm round glass coverslips and incubated in $Ca^{2+}/Mg^{2+}$ supplemented HEPES buffer (HBSS, Invitrogen) were transferred onto the stage of an Olympus IX81 inverted microscope (Melville, NY) equipped with a CellR imaging system and a beam-splitter optical device (Multispec Microimager; Optical Insights). Images were acquired using a 40×, 1.4 NA objective (Olympus) and the CellR software. Analysis of fluorescence was performed following background subtraction over mitochondrial regions of interests (ROIs), using the multi-measure plug-in of Image J (NIH). Representative still frames are pseudocolor coded.

For real time imaging of mitochondrial ΔpH, cells ($5 \times 10^4$) seeded onto 24-mm round glass coverslips were transfected with pSypHer-dMito[34] and analyzed after 24 or 48 h after transfection to express Cre recombinase. Ratiometric images of the 535-emission fluorescence were acquired every 10 s by alternate excitation of cells at 430 and 500 nm for 100 ms. Mean fluorescence ratios of selected ROIs matching mitochondria were measured and expressed as mtSypHer (430/500 nm) ratios.

Mitochondrial ATP content was determined by FRET image analysis of cells transfected with pcDNA-ATeam1.03[38]. Sequential images of the 525 and 475 nm fluorescence emission after alternate excitation at 435 nm for 100 ms were acquired every 30 s.

For TMRM fluorescence analysis, sequential images were acquired every 30 s[11].

**Transmission electron microscopy**. Electron microscopy (EM) imaging of cells was performed as described[26] on a Tecnai G2 (FEI) transmission electron microscope operating at 100 kV. Images were captured using a Veleta (Olympus Imaging System) digital camera (pixel size 13 × 13 μm; pixel size at a 46,000X magnification with screen magnification of 3 × 0.1 × 0.1 nm). For morphometric analysis of mitochondrial cristae, performed in a blind fashion on at least five mitochondria/cell from six randomly selected cells ($n = 3$ independent experiments), maximal cristae width was measured using the ImageJ Multimeasure plug-in[3]; the number of horizontal cristae and cristae junctions were quantified manually.

**Immunoblotting**. For SDS-PAGE experiments, proteins (30–40 μg) were separated on Any-KD (BioRad) polyacrilamide precasted gels, transferred onto PVDF membranes (BioRad) and probed using the indicated antibodies and isotype matched HRP-conjugated secondary antibodies. The following primary antibodies were employed at 1:1000 dilution: OPA1 (BD, #612607), ATP5A (ab14748), ATP5B (ab14705), IF1 (ab110277), SDHA (ab14715), ATP5 subunit e (ab54879) from Abcam; ACTIN (Chemicon) was used at a 1:30000 dilution (MAB1501 Millipore). Densitometry was performed using ImageJ gel measure tool and analyzing the optical density of selected ROIs containing ATP synthase dimers, monomers and F1. Uncropped scans of relevant blots are included in Supplementary Figure 9.

**Isolated mitochondria assays**. Mitochondria were extracted from cells grown in 500 cm$^2$ dishes[67]. After isolation, mitochondrial protein concentration was determined by Bradford assay (BioRad) and 0.5 mg/ml protein were incubated in Experimental Buffer (EB: 150 mM KCl, 10 mM Tris MOPS, 10 μM EGTA-Tris, 1 mM KHPO4, 5 mM glutamate, 2.5 mM malate) and further incubated where indicated with 10 pmol/mg cBID for 30 min at room temperature.

**Blue-native polyacrylamide gel electrophoresis**. Isolated mitochondria were resuspended (0.5 mg/ml) in NativePAGE Sample buffer (Invitrogen) containing 1.1% (w/V) digitonin (Sigma) and protease-inhibitor cocktail (Sigma). After 5 min on ice, the lysate was spun at 20,000xg for 30 min at 4 °C. G250 (5%, 1 μl/100 μg protein, Invitrogen) was added to the supernatant 30–40 μg of protein were loaded onto a 3–12% native precast gel (Invitrogen).

**Protein crosslinking**. Where indicated, mitochondrial extracts were crosslinked in the presence of the zero-length crosslinker EDC[11]. Proteins (50 μg) were incubated for 30 min at 37 °C in PBS supplemented with 10 mM EDC. The reaction was quenched by adding 15 mM dithiothreitol (DTT) and proteins were separated by SDS-PAGE after 15 min.

**Immunoprecipitation**. Isolated mitochondria were lysed in 150 mM NaCl, 25 mM Tris-Cl pH 7.4, 1 mM EDTA, 5% glycerol, 0.1% Triton X-100 in the presence of Protease Inhibitory Cocktail (PIC) (Sigma). Lysates (250 μg) were precleared on 20 μl of Protein-A agarose beads (Roche) for 30 min at 4 °C and subsequently immunoprecipitated with protein-A agarose beads coupled with the indicated antibodies in lysis buffer overnight at 4 °C[26]. The immunoprecipitated material was separated by SDS-PAGE.

**Liposomes preparation**. Liposomes were prepared from purified soybean asolectin (L-α-phosphatidylcholine, Sigma). Lipids were dissolved in chloroform (5 mg/ml) until a homogeneous mixture was obtained; the solvent was then evaporated on a nitrogen stream to yield a thin lipids layer on a glass tube bottom. The lipid film was thoroughly dried by placing the tube on a vacuum pump overnight to remove residual chloroform. To obtain large multilamellar vesicles (LMV) the lipid film was hydrated with 1 mL buffered solution (150 mM KCl, 10 mM Hepes, pH 7.4), containing where indicated 20 μg of purified recombinant OPA1 (rOPA1), and gently agitated at room temperature. When rOPA1 was added, liposomes were then centrifuged at 30,000xg for 5 min and resuspended in fresh buffer devoid of rOPA1. LMV were then downsized to liposomes (large unilamellar vesicles, LUV) by extrusion through a polycarbonate filter with a pore size of 100 nm (Avanti Polar Lipids).

Intact mammalian F1FO-ATP synthase was purified from beef heart mitochondria as described[68] and inserted into freshly made liposomes by direct incubation of the protein (20 μg) with the liposomes solution for 30 min at 4 °C. To remove non-inserted F1FO-ATP synthase complexes proteoliposomes were pelleted by centrifugation at 30,000xg for 5 min and resuspended in buffer.

**Negative staining EM**. Twenty-five μl of containing freshly prepared proteoliposomes suspension were placed on a 400-mesh holey film grid, stained with 1% $UO_2(CH_3COO)_2$ and observed with a Tecnai G2 (FEI) transmission electron microscope operating at 100 kV. Images were captured using a Veleta (Olympus Imaging System) digital camera.

**Blue native PAGE and silver staining of recombinant proteins**. Purified ATP synthase was supplemented with Coomassie Blue G-250 (Serva) and applied to 1D 3–12% polyacrylamide gradient BNE (Invitrogen). After electrophoresis, proteins were eluted from BNE gels overnight as follows. Bands corresponding to monomers of F-ATP synthase were excised and diluted with 25 mM tricine, 7.5 mM Bis-Tris, and 1% (w/V) n-heptyl β-D-thioglucopyranoside, pH 7.0 supplemented with 8 mM ATP-Tris, and 10 mM MgSO4. Samples were incubated at 4 °C, centrifuged at 20,000 × g for 20 min at 4 °C, and supernatants were inserted into liposomes as described above.

Liposomes were then solubilized with 4% (wt/vol) freshly prepared digitonin, supplemented with Coomassie Blue G-250 (Serva), and loaded in 2D-BNE followed by silver staining. Gels were fixed overnight with formaldehyde, rinsed with ethanol, and pretreated with a solution of 0.8 mM Na2S2O3; gels were then stained with 11.2 mM AgNO3 for 20 min, and then with 0.6 M Na2CO3 for the time required for the bands to be revealed.

**Production and purification of recombinant OPA1**. OPA1 mouse transcript variant 2 (NM_133752) was amplified from position 502 to the stop codon to produce recombinant OPA1 (rOPA1) lacking the N-terminal transmembrane domain. The PCR product was cloned into pET21 + (Novagen) which adds a His-tag to the C-terminus of the encoded protein and expressed in E. coli. Protein production was induced with 0.5 μM IPTG for 20 h at 18 °C. E. coli were collected by centrifugation and the pellet was resuspended in lysis buffer (40 mM Hepes/KOH, 500 mM NaCl, 10% glycerol, 5 mM MgCl2, 5 mM β-mercaptoethanol, 0.5% Triton-X-100, 2% Tween-20, 1 mM PMSF, 20 mM Imidazole, Roche protease-inhibitor cocktail, pH 8.0). Cells were lysed by sonication and cell debris was removed by centrifugation at 14,000xg for 45 min, 4 °C. rOPA1 was purified by Ni-NTA batch chromatography. Prior to elution, beads were washed with 1 mM ATP and 10 mM MgCl2 in lysis buffer for 30 min at room temperature followed by an intermediate washing step in wash buffer (40 mM Hepes/KOH, 300 mM NaCl, 0.05% Triton-X-100, 5 mM MgCl2, 5 mM β-mercaptoethanol, 10% glycerol, 20 mM Imidazole, pH 8.0). rOPA1 was eluted with increasing concentrations of imidazole in elution buffer (40 mM Hepes/KOH, 0.05% Triton-X-100, 5 mM β-mercaptoethanol, 10% glycerol, pH 7.4). Imidazole was removed by dialysis and the protein was concentrated in storage buffer (40 mM Hepes/KOH, 0.05% Triton-X-100, 0.3 mM TCEP, 10% glycerol) and stored at -80 °C until use.

**In-gel ATPase activity assay**. ATPase in-gel activity was measured directly in the gel by incubating it for 2 h in a solution containing 35 mM Tris, 270 mM glycine, 14 mM MgSO4, 0.2% Pb(NO3)2, and 8 mM ATP, pH 7.8[69,70].

**BNGE based semi-quantitative proteomic analysis**. Mass spectrometry analysis of mitochondrial complexes from mouse CD1 hearts was performed as indicated[26]. False discovery rate (FDR) of identification was controlled as described by the algorithm Dxtractor. Median values of all identified ATP synthase subunits and representative color-contour plots of spectral counts of OPA1 peptides, core F1 ATP synthase subunits and FO complex subunit e required for dimerization were considered for analysis.

**Cell death assays**. For cell death analysis, $3.5 \times 10^3$ cells/cm$^2$ of the indicated genotype were co-transfected with SureSilencing® shRNA and pIRES2-eGFP plasmids (3:1 ratio). After 48 h, the medium was replaced with galactose-supplemented DMEM and after 24 h cells were treated with 5 μM antimycin A for 6 h. Cell death was assessed by flow cytometry detection (FACSCalibur) of double Annexin-V-APC/PI positive events from the transfected GFP+ cell population[3].

**Statistical analysis**. Results are expressed as the mean ± SEM values of the indicated number ($n$) of independent experiments. Individual data points are overlaid on the bar graphs. Statistical significance was determined by Student's $t$ test or ANOVA between the indicated samples. $P$ values are indicated in the legends and $P < 0.05$ was considered significant.

**Data availability**. The data that support the findings of this study are available from the corresponding author upon reasonable request. The uncropped blots can be found in Supplementary Figure 9.

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

## Acknowledgements

The authors thank Drs. F. Caicci and E. Boldrin (Department of Biology, University of Padova) for EM sample preparation; Drs. N. Demaurex (University of Geneva, Switzerland), H. Imamura and H. Koji (University of Kyoto, Japan) for reagents, J.P. Bolaños and A. Almeida (University of Salamanca, Spain) for facilities and discussion. R.Q.-C. was supported by an AIRC Postdoctoral Fellowship, a Fondazione Umberto Veronesi Postdoctoral Fellowship and is currently a recipient of a Juan de la Cierva-Incorporación fellowship from the Spanish Ministry of Economy, Industry and Competitiveness (IJCI-2015–26225). This work was supported by Telethon-Italy GPP10005, GGP14187, GGP15091; AIRC Italy IG-15748, ERC FP7-282280, FP7 CIG PCIG13-GA-2013-618697; Italian Ministry of Research FIRB RBAP11Z3YA_005 to L.S. C.Ge. is supported by JST, CREST Grant JPMJCR13M4 (to Genij Kurisu and C.Ge.), the Platform for Drug Design, Discovery and Development from MEXT, Japan and the Grants-in-Aid for Scientific Research (Kiban B: 17H03647) from MEXT, Japan. JAE is supported by Spanish Ministry of Economy, Industry and Competitiveness (SAF2015-65633-R; SAF2015-71521-REDC). The CNIC is supported by MINECO and Pro-CNIC Foundation and is a SO-MINECO (award SEV-2015-0505).

## Author contributions

R.Q.-C. and L.S. conceived the project, acquired funds and wrote the manuscript. R.Q.-C., C.Gl., M.C., C.Q., A.P., A.U., and M.E.S performed and analyzed experiments. E.C. and M.E.S. wrote software and analyzed data; J.V., J.A.E., C.Ge., P.B, M.E.S. provided reagents, conceptualized experiments and edited the manuscript; L.S. supervised the project.

## Additional information

**Competing interests:** The authors declare no competing interests.

