## [Peer Review File · Nature Communications]

Reviewers' comments:

Reviewer #1 (Remarks to the Author):

Safeguarding the peculiar ultrastructure of mitochondria is of fundamental importance for the functionality of these organelles and thus cellular energy homeostasis. Maintenance and dynamic adaptation of cristae membranes, which represent the main sites of oxidative phosphorylation in mitochondria-containing cells, is a hallmark of this process. Several protein machineries affecting cristae shape and remodeling have been identified during the last years, but little is known about how they cooperate. In this manuscript, Scorrano and colleagues analyze the relationship between two major determinants of cristae architecture: oligomeric OPA1 and ATP synthase complexes. They find that OPA1 levels and apoptotic cristae remodeling affect ATP synthase dimerization. OPA1 and ATP synthase dimerization are required to maintain reverse (proton-pumping) activity of the ATP synthase (ATPase) under conditions of respiratory chain impairment. The authors conclude that OPA1 protects against mitochondrial dysfunction at least in part by supporting ATP synthase dimerization and function.

The work sheds new light on the important question how mitochondrial ultrastructure is controlled to assure the functional adaptations of mitochondria required upon metabolic adaptation and exposure to stressors. In principle, this paper will be of great interest for the broad readership of Nature Communications. However, a the following problems urgently need to be addressed prior to publication.

Major scientific points:

- Cristae morphogenesis depends on the formation of extended rows of ATP synthase dimers. Thus, with respect to cristae remodeling ATP synthase oligomerization rather than dimerization is the decisive parameter. It has been shown by other labs that the levels of MICOS complex core components affect the formation of higher-order ATP synthase oligomers (but not dimers). Under many of the conditions tested in this work monomer-to-dimer ratios are only moderately changed, but strong conclusions are derived. It is therefore necessary to experimentally address the effects of OPA1 on the formation of higher-order ATP synthase oligomers.
- Do the authors conclude that OPA1 affects ATP synthase dimerization solely via its membrane-shaping activity, i.e. control of cristae width and tightening of crista junctions? It is unclear to me, how this should work. Do the authors suggest that cristae width is the superior parameter that determines curvature at tips and rims? Again such changes should rather relate to the oligomerization of ATP synthase dimers rather than the dimerization per se. Is there any evidence for a direct interaction of (dimer/oligomer-promoting) ATP synthase components and OPA1?
- There is substantial evidence for an autonomous membrane-shaping activity of ATP synthase oligomers independent of OPA1. As the authors present their data in way that OPA1 appears to be the active part in the relationship with ATP synthase, it will be

important to directly test, if alterations in ATP synthase dimer/oligomer levels (e.g. ATP5K knock-down or overexpression) affect OPA1 oligomerization. If this is the case, mutual regulation of both machineries appears more likely than an unidirectional process with OPA1 being the trigger.

- Overall the paper is very difficult to read. The authors should better explain their assays and rationales to improve accessibility especially for non-expert readers. In particular, the chapter "Opa1 requires ATPase function to maintain mitochondrial function" (page 8f.) is hard to follow. What do the authors for example mean by the sentence on the differences between TMRM and mtSypHer assays (lane 164ff.)? Moreover, it is important to show that the decrease in mtATeam ratios upon addition of Antimycin A is indeed oligomycin-sensitive and may therefore unambiguously be assigned to the proton-pumping (i.e. coupled) ATP synthase under the analyzed conditions.

- Page 12/13, lines 272-274: In the last part of the Discussion the authors very rightly elaborate on a possible role of the ATPase inhibitor IF1 in linking OPA1 function to the proton-pumping (reverse) ATP synthase/ATPase activity. This is an extremely interesting aspect (however, I do not really understand the mechanistic explanation that is suggested). Therefore, the informative value of this manuscript would benefit a lot from an experimental analysis of the behavior of IF1 under the relevant conditions with blocked respiratory chain and overexpressed or ablated OPA1.

Further scientific and technical points:

- Figure 1b/c: The amounts of large OPA1 oligomers in untreated and cBID-treated mitochondria in Figure 1b look very similar. Although the loss of OPA1 oligomers under these conditions is documented again in Figure 1c (and has been extensively studied in the past), still a representative and clear Western blot image should be shown.

- Page 5, line 99f.: cBID treatment of mitochondria lowers the amount of ATP synthase dimers as shown by BN-PAGE analysis, which is reflected by the loss of the dimerization-promoting subunit ATP5K. Because the overall amount of ATP5K in the gel is much lower upon cBID treatment: What happens to the protein under these conditions (in vitro treatment with cBID)? Does it dissociate into smaller protein complexes not visible in Figure 1e? Or is the protein degraded (in vitro)?

- Figure 1f: How many times was this experiment repeated? Error bars should be shown. Moreover, dimer-to-monomer ratios will be the better unit for the y-axis instead of total amounts of dimer counts. Do the authors have any idea, why the amount of ATP5E (epsilon subunit) in the ATP synthase dimer region of the gel is rather increased?

- Figure 2e/f: The relative increase of ATPase activity in the dimer bands in Opa1tg mitochondria is not visible in Figure 2e. Is the quantified difference in Figure 1f significant?

- Figure 2g: Why are the overall amounts of ATP synthase complexes in the mitochondria so

different between the samples shown? (The treatment with cBID and variants are done in vitro with isolated mitochondria.) This makes it again very difficult to appreciate ratio differences from the visual inspection of the Western blot. Again, are the quantified differences in Figure 2h/i significant?

- Page 6, line 116 ff. and Figure S1c: Why are the total levels of ATP synthase so much reduced in OPA1-deleted MEFs? There is hardly any ATPase activity detected in the gel regions where ATP synthase monomers and dimers are expected. How was a monomer/dimer ratio calculated for these samples?

- Figure 3b: There is a problem with the labeling. I assume the red bars represent mtSypHer ratios in Opa1tg mitochondria.

- I assume that Figure S4a can be considered as "oligomycin only"-control for the experiments in Figure 5c/d. Such controls are also necessary for Figure 5a/b (mtSypHer ratio in WT and Opa1tg mitochondria) and for figure 5e/f (mtSypHer ratios in WT and OPA1-deleted MEFs).

- Page 9/10, lines 203 - 206: The logic of this statement is not clear to me? Why do the results support this idea? There could be other reasons for the observed effects.

- Page 10, line 218: This sentence refers to Figure 7f (not 6f as indicated).

Reviewer #2 (Remarks to the Author):

Overall, this paper is well done and the concept is novel. The take-home message is a bit confusing to the reader, is it OPA1 that affects ATPase dimers directly or is this the result of OPA1 effects on cristae morphology? As the mechanism as to how OPA1 mediates these effects is unclear, the manuscript would be significantly strengthened if this point could be better addressed. Below are my other comments.

1) In figure 2, the authors should provide data to demonstrate that knockout OPA1 does not affect mtDNA or translation. This is mentioned but should also be shown.

2) Many conditions used disrupt cristae and OPA1 together, thus it is unclear whether the effects shown are directly due to OPA1, or secondary effects due to loss of cristae structure in the absence of OPA1. Could the authors find a way to disrupt cristae without altering OPA1? What would happen to the ATP synthase in that case?

3) In figure 3 the authors perform a series of complex experiments using matrix pH as a parameter of mitochondrial function. It would also be useful to look at mitochondrial ATP production and oxygen consumption, given that this paper is linking this to ATP synthase.

4) In figure 4 and 5 the authors discuss how OPA1 mediated protection and maintenance of

mitochondria function requires ATPase activity. The authors should show what happens to OPA1 HMWC (high molecular weight complexes) in the absence of ATPsynthase activity?

5) Why are there no changes in basal ATP content in OPA1tg (Figure 5) when in Figure 2 it shows that there is increased ATPase activity in OPA1tg? This should be explained.

6) There should be more discussion on how OPA1 affects ATPsynthase dimerization. Is it by OPA1 directly or is it indirectly through OPA1-mediated effects on cristae architecture. As the mechanism by which this occurs is unknown, the authors should try to clarify this point.

Reviewer #3 (Remarks to the Author):

Qunintana-Cabrera et al., has investigated the protective role of OPA1 against apoptosis induced mitochondrial dysfunction. Using a variety of BID mutants which dissect the process of apoptosis into outer membrane permeabilization and/or cristae remodelling, Qunintana-Cabrera demonstrates that:

- 1) OPA1 is important for the stability of ATP synthase complexes
- 2) OPA1 overexpression protects against BID induced ATP synthase destabilisation,
- 3) OPA1 overexpression prevents matrix acidification and membrane potential loss caused by complex III inhibition. (A process requiring active ATP synthase dimers)
- 4) OPA1 overexpression counteracts changes in cristae morphology caused by reduction in concentration of ATP synthase dimers.
- 5) Cytochrome c release requires permeabilisation of the outer mitochondrial membrane but not cristae remodelling.

From these results, the authors predict that OPA1 protects against apoptosis by promoting ATP synthase dimerization and 'reversal activity' (e.g. ATP hydrolysis).

Main comments:

The manuscript in its current state is inaccessible to the general readership of Nature communications. The authors have a strong tendency to state a hypothesis, reference a figure and then draw a conclusion without properly describing the results being presented. This means the reader needs to be an expert in the techniques used by the authors in order to understand what has lead the authors to draw their conclusions. In addition, because the main figure legends lack sufficient details, it is impossible to understand the figures without the text or the text without the figures.

Another major issue with the manuscript is the significance of the results discussed. The authors do not discuss the degree of change required to make a result significant. E.g. How

significant is a 5nm change in cristae width given the pixel size used for imaging and the technique used (Fig 2b)? Is a difference of 0.006 a.u./min really a substantial change (Fig 5i)? Is a difference of 1 spectral count significant or in the realms of noise (Fig 1f)?

In the principle conclusions of the manuscript the authors state that OPA1 promotes “reversal activity” of the ATP synthase. More concrete evidence is required to support this claim, and also evidence to indicate the effect of OPA1 overexpression on the IF1 protein whose function is to prevent ATP hydrolysis by the ATP synthase in mammalian cells. In the authors discussion, they suggest OPA1 interferes with the function of IF1 but the two proteins are in different compartments of the mitochondria. How would this work?

In addition, it is not clear that OPA1 promotes ATP synthase dimerization from the data presented. I agree that the authors present clear data demonstrating that active ATP synthase dimers are required for overexpressed OPA1 to protect against matrix acidification and membrane potential loss caused by complex III inhibition but there is no evidence to say that it actively promotes ATP synthase dimerization or how this might occur.

Point-by-point comments

Why use ATPase to refer to the mitochondrial ATP synthase. In cells, the main function of the ATPase is ATP synthesis not hydrolysis (hence ATP synthase).

Title: The data presented demonstrates that the protective effect of OPA1 on mitochondrial function requires active ATP synthase dimers. This should be reflected more accurately in the title. E.g. ‘... requires active mitochondria ATP synthase dimers’ instead of ‘... requires mitochondrial ATP synthase dimerization’

Abstract: I do not believe the data presented provides sufficient evidence to support the notion that OPA1 stimulates ATPase dimerization. In addition, the data present does not show apoptotic and genetic manipulation based changes to cristae shape. This is inferred from previous publication and should be removed from the abstract. Instead the abstract should be rewritten to focus on the concrete results of the paper as stated in the title.

Introduction:

Line 42: Ref 11 and 12 are duplications

Line 77-79: I do not think the data presented provides sufficient reliable evidence to support the statement “OPA1 promotes ATPase dimerization and reversal activity’. Also I did not encounter a clear explanation or presentation of the mechanisms through “which cristae stabilization by OPA1 protects from primary mitochondrial dysfunction”. In addition, what do the authors mean by “primary mitochondrial dysfunction.”?

Results section “ATPase dimers are destabilized during cristae remodelling”

Line 86: Purified heart mitochondria was obtained from which animal?
Wildtype/overexpression?

Line 91: therefrom should be changed to henceforth

Line 98: "Fig. 1c) revealed that cristae remodelling resulted in the expected disassembly of HMW OPA1-containing complexes".

There seems to be a discrepancy between the complexome and western blot images. In fig 1b, OPA1 appears to be present at roughly the same level as in the untreated mitochondria but in the complexome (fig 1c) there appears to be a large difference. This needs to be explained.

Line 95: Why use the gene name for the subunits of the ATP synthase when you are detecting the protein?

Line 96-97: "were reduced during cristae remodelling in the BNGE regions corresponding to ATPase dimers (Fig 1d)". The region on the gel corresponding to the ATPase dimers need to be indicated for the reader.

Line 102: " the spectral counts remained ~72% of the untreated (Fig 1e)" and Fig 1F.

The graph for subunit e shows a difference of one spectral count between BID and BID^{KAAA} treated mitochondria. When is a change in spectral count considered significant?

Why does the spectral count for ATP5E triple in the BID treated mitochondria?

Why does the spectral count stay the same between BID and BID^{KKAA} treated mitochondria for the ATP5L?

The spectral counts for ATP5L and ATP5E are in the same range as ATP5K. How reliable are these results and how much can you conclude from these data?

Results section "ATPase stability is regulated by Opa1 levels"

Line 110: more explanation as to why each stage is performed would make the manuscript easier to read. E.g. "Because OPA1 overexpression stabilizes mitochondrial RCS, we verified if the same was also true for the ATPase. For these experiments we used adult mouse fibroblasts (MAFs) derived from transgenic mice which were overexpressing OPA1 (Opa1^{tg}). In these cells, OPA1 was expressed 1.5-fold greater than the WT and had narrower gaps in the cristae. ..."

Line 112 & Fig 2A: How was the 1.5-fold increase in Opa1 quantified?

Line 112 & Fig 2B: How significant is the 5 nm narrowing of the cristae especially given the pixel size, camera resolution and fixation procedure? How many different mitochondria were analysed from how many cells? Were these from the same fixation experiment or different? Could the change in distance reflect different angled slices through the lamellar cristae or even different amounts of dehydration between different cells during the fixation/embedding procedure?

Line 133 & Fig 2d-f: What evidence leads you to the conclusion that OPA1 over expression increases the stability of the F1Fo ATP synthase dimer? Why is figure S1c in the supplementary?

Line 115: How do you know that you didn't alter mtDNA content or translation caused by chronically impaired fusion? Could mtDNA content or translation be modified other than by chronically impaired fusion? Also what exactly does 'chronically impaired fusion' mean? Are you referring to cell-cell fusion or mitochondrial fusion?

Line 124: "cBID induced complete cytochrome c release in WT but not Opa1^{tg} MAFs"
What indicates complete release?

Line 126: "(iii) cytochrome c release was comparably reduced when we treated WT cells with cBID^{KKAA} or Opa1^{tg} cells with cBID". How do you define "comparably reduced"? What are you comparing the results with? Why is this data in the supplementary figures when you discuss it at length in the main text?

Line 127: "Thus, OPA1-controlled cristae remodelling is required for complete cytochrome c release also in permeabilized cells". Is this conclusion valid? As described by the authors, cBID^{KKAA} permeabilises the outer membrane but does not cause cristae remodelling (Line 89) and cBID^{G94E} causes cristae remodelling but without permeabilizing the outer mitochondrial membrane (line 121). Fig S2 indicates cytochrome c is released using cBID and cBID^{KKAA} but not cBID^{G94E} indicating cytochrome c release requires outer membrane permeabilisation but not cristae remodelling.

Lines 122-128: Why did the authors choose to place the graphs for this section as supplementary and not main text?

Line 132 & fig 2g-i: Why is the figure labelled with BID and not cBID? What is the difference?

Line 133: "Thus in chronically or acutely remodelled cristae ATPase dimers are destabilized"
What generates chronically remodelled cristae and what produces acutely remodelled cristae? What is the difference between chronic and acutely remodelled cristae? Do you have data to support these assignments?

The only conclusions that can be drawn from this section is that 1) ATP synthase stability decreases with cristae remodelling and 2) overexpression of OPA1 appears to protect against BID induced ATP synthases destabilisation.

Line 133: Does cBID G94E and cBID cause cristae remodelling in Opa1^{tg}? Please provide evidence for this.

Results section "OPA1 counteracts mitochondrial dysfunction caused by cristae remodelling"

Line 139: "Matrix pH was comparable in WT and Opa1^{tg} MAFs". How are they comparable?

Line 146. Corresponding figures need to be referred to at beginning of explanation.

Line 151: Shouldn't tBID^{G94A} be tBID^{G94E}?

Line 153: Δ pH should be replaced with matrix acidification as this is what the authors are measuring.

Line 153: “induced by the tBID mutants that cause cristae remodelling”. Only one mutant affects cristae remodelling (tBID^{G94E}). Also the protection offered by OPA1 overexpression seems to be equally strong in all tBID variants not just the one that effects cristae remodelling (fig 3a). The authors must provide addition proof that the protection offered by OPA1 overexpression has more effect on the mutant which only remodels the cristae.

Fig 3b +AA. What change in ratio indicates a significant difference? Is a 0.05 change in ratio really significant? What does the “ratio” measure?

Line 154: conclusion should represent the conditions tested. E.g. “OPA1 overexpression protects cells against matrix acidification caused by inhibition of complex III”
The section title should also be modified to reflect the data discussed e.g. OPA1 overexpression protects again matrix acidification caused by complex III inhibition.

Results section “Opa1 requires ATPase activity to maintain mitochondrial function”

Line 163: What does “fully protected” mean?

Line 169: “Mitochondrial dysfunction” is a broad term. Authors should clarify which type of “mitochondrial dysfunction” they are referring to?

Line 172: Can the authors really say that OPA1 requires mitochondrial ATPase activity to sustain Δ pH upon CIII inhibition? Isn't it more likely that oligomycin inhibition of ATP synthase overrides (is dominant to) complex III inhibition by Antimycin A? If ATP synthase function is inhibited, the respiratory chain complexes also become inhibited due to a lack of dissipation of the intra-cristae proton concentration. The authors need to discuss this. A better conclusion would be: “the protection OPA1 overexpression provides on matrix acidification caused by complex III inhibition requires active ATP synthases”

Line 175: The ATPase activity of F-type ATP synthases in cells (especially mammals) is minimal because as the matrix acidifies and the conditions become appropriate for ATP hydrolysis by the ATP synthase, the inhibitor factor (IF1) binds to the ATP synthase preventing ATP hydrolysis.

Line 178: Oligomycin binds to the membrane embedded rotor ring of ATP synthase preventing the movement of protons across the membrane. Thus in oligomycin inhibited cells there is no loss in membrane potential or Δ pH. This paragraph should be removed or modified to reflect this knowledge.

Line 185: The function of the IF1 protein in mammalian mitochondria is to inhibit ATP hydrolysis by ATP synthase when the matrix becomes acidic. How does OPA1 overexpression and OPA1 knockdowns effect matrix IF1 concentration?

Line 185 & Fig 5g-i. How significant is a change of 0.06 a.u./min (fig 5i) or a mtATeam ratio change of 1% (Fig 5h)?

Line 186: "In conclusion, OPA1 stimulates F1Fo-ATP synthase reversal activity to sustain mitochondrial ΔpH ". The authors do not provide any evidence to support this claim. They at least need to validate the effect of OPA1 on matrix IF1 concentration and the significance of a 0.006 a.u./min change in mtATeam rate.

Line 194: Please change ATPase to ATPase.

Line 199: "While ATP5k downregulation did not affect cristae width". This is not possible to accurately measure with the technique used.

Line 219: Please change "blunt" to curtail.

Results section "OPA1 requires ATPase dimerization to maintain mitochondrial function"

Line 189: "In Opa1^{tg}", this should be changed to "In mitochondria from Opa1 overexpression MAFs".

Line 189: "cristae are narrower" This is difficult to quantify or validate with the technique used.

Line 189: "ATPase is more dimerized and active". The data presented in the manuscript provides no evidence for more dimers only that they are more stable (less likely to disassemble)

Line 191: "mitochondrial dysfunction is a too broader term. Please be more specific by describing the type of mitochondrial dysfunction you have analysed or are going to test.

Discussion

Line 230, 235: It is not clear what the authors means by "ameliorates" and "amelioration". Please exchange for a simpler word. This would make the statements more accessible to readers where English is not their first language.

Line 272: My understanding is that OPA1 is an intra-crista space protein which in some form binds to the inner mitochondrial membrane. IF1 is a soluble matrix protein. How can overexpression of OPA1 prevent binding of the IF1 to the ATP synthase when they are not located in the same compartment?

Figures:

Figure 1:

- 1) Please indicate location of ATP synthase dimer and monomer band.
- 2) Please complete parenthesis in figure 1F
- 3) What does (--) mean? Please add to figure legend description.
- 4) Why are figures 1a-e labelled cBID and cBID^{KKAA} and fig. 1F is labelled BID^{WT} and BID^{KKAA}? What is the difference.
- 5) What does the color number scale refer to?
- 6) What does the numbers on the left and right (fig 1c) refer to?

Supplementary figure 1:

- 1) What is the staining used in the left image of S1C?
- 2) Fig S1C, left label ATPA should be changed to ATPase.

Figure 2:

- 1) Fig 1c, y-axis + bars should have a break to indicate scale does not extend from zero (see Fig 1h)
- 2) Fig 1d, please change the label ATPA to ATPase to make it consistent with text. Also add label to figure legend to improve clarity

Figures 3-7:

- 1) What do the color bar mean in the first fluorescent image of each panel? What is the meaning of the scale that is indicated?
- 2) AA, EV and FCCP need to be defined in the figure legend.
- 3) shSCR and shATP5k need to be explained in figure legend.
- 4) What is being tested in each image?

Methods:

Line 364: F₀. 0 should be o as it stands for oligomycin.

General response to reviewers' comments:

We thank the reviewers for their time and attention. Their very insightful comments helped improving our manuscript and we hope that now it can be acceptable for publication.

All reviewers recognized the quality and novelty ("*this paper is well done and the concept is novel*". Reviewer #2) as well as the relevance of the work ("*The work sheds new light on the important question how mitochondrial ultrastructure is controlled to assure the functional adaptations of mitochondria*. Reviewer #1) and its suitability for "*the broad readership of Nature Communications*" (Reviewer #1).

We agree with the referees on the need for a more accessible writing for "*non-expert readers*" (Reviewer #1) and the "*general readership of Nature communications*" (Reviewer #3). We have therefore turned around the order of the figures to improve the clarity of the presentation and largely rewrote it; we think it flows much better now.

The major critique on the conclusions of the work was to address whether Opa1 exerted the ATP synthase dependent protection by a direct interaction with the latter. We hope that the reviewers can appreciate how seriously we took this concern and that we really ran the extra mile to verify whether Opa1 directly stimulates ATP synthase oligomerization.

First, we performed immunoprecipitation, BNGE and crosslinking analyses that revealed an interaction between ATPase and Opa1, confirming the former results by Patten et al¹ who described the presence of ATPase subunits within Opa1 putative interactors by mass spectroscopy. However, that the two proteins can directly interact does not mean that Opa1 stimulates ATP synthase oligomerization observed in *Opa1^{tg}* cells. To test the functional consequences of the model of direct Opa1-ATP synthase interaction, we joined forces with Dr. Christoph Gerle, a leading expert on ATP synthase purification and structure and we setup an in vitro system of reconstituted liposomes with recombinant Opa1 and with highly purified ATP synthase. In this setting, Opa1 did not promote ATPase oligomerization (please see **new** Fig. 5). Together with other experiments of epistatic analysis of the Opa1-ATP synthase pathway in cristae morphology, we conclude that Opa1 favors ATP synthase oligomerization indirectly, by acting on cristae shape. These new experiments reinforce the final message of our paper: Opa1 promotes ATP synthase oligomerization to sustain membrane potential and cell viability when respiratory chain is inhibited.

Reviewer #1 (Remarks to the Author):

Safeguarding the peculiar ultrastructure of mitochondria is of fundamental importance for the functionality of these organelles and thus cellular energy homeostasis. Maintenance and dynamic adaptation of cristae membranes, which represent the main sites of oxidative phosphorylation in mitochondria-containing cells, is a hallmark of this process. Several protein machineries affecting cristae shape and remodeling have been identified during the last years, but little is known about how they cooperate. In this manuscript, Scorrano and colleagues analyze the relationship between two major determinants of cristae architecture: oligomeric OPA1 and ATP synthase complexes. They find that OPA1 levels and apoptotic cristae remodeling affect ATP synthase dimerization. OPA1 and ATP synthase dimerization are required to maintain reverse (proton-pumping) activity of the ATP synthase (ATPase) under conditions of respiratory chain impairment. The authors conclude that OPA1 protects against mitochondrial dysfunction at least in part by supporting ATP synthase dimerization and function. The work sheds new light on the important question how mitochondrial ultrastructure is controlled to assure the functional adaptations of mitochondria required upon metabolic adaptation and exposure to stressors. In principle, this paper will be of great interest for the broad readership of Nature Communications. However, the following problems urgently need to be addressed prior to publication.

We would like to thank the reviewer for the appreciation of our work.

Major scientific points:

Cristae morphogenesis depends on the formation of extended rows of ATP synthase dimers. Thus, with respect to cristae remodeling ATP synthase oligomerization rather than dimerization is the decisive parameter. It has been shown by other labs that the levels of MICOS complex core components affect the formation of higher-order ATP synthase oligomers (but not dimers). Under many of the conditions tested in this work monomer-to-dimer ratios are only moderately changed, but strong conclusions are derived. It is therefore necessary to experimentally address the effects of OPA1 on the formation of higher-order ATP synthase oligomers.

Thank you for the insightful suggestions. To address this comment, we performed the BNGE to include ATP synthase oligomers and we have recalculated and plotted the composition and activity of ATPase from all experiments as ratios oligomers vs. total ATPase, i.e. $\text{sum of (oligomers + dimers) / (oligomers + dimers + monomers + F}_1\text{)}$. The results support the ability of Opa1 to promote the stabilization of oligomeric, not only dimeric forms of ATPase. These **new** experiments can be found in Figs. 3, 5, 6

Do the authors conclude that OPA1 affects ATP synthase dimerization solely via its membrane-shaping activity, i.e. control of cristae width and tightening of crista junctions? It is

unclear to me, how this should work. Do the authors suggest that cristae width is the superior parameter that determines curvature at tips and rims? Again such changes should rather relate to the oligomerization of ATP synthase dimers rather than the dimerization per se. Is there any evidence for a direct interaction of (dimer/oligomer-promoting) ATP synthase components and OPA1?

This is a great question. In our previous work, we provided evidence that cristae membrane curvature is sufficient to determine stability and assembly of respiratory complexes (RCC) into supercomplexes (RCS)². Here we show that (i) OPA1 promotes ATP synthase oligomerization; (ii) OPA1 maintains cristae tight independently of ATP synthase, because cristae are narrow in Opa1^{tg} cells irrespective of ATP5k silencing that reduces ATP synthase oligomerization. Thus, there was already circumstantial evidence that membrane curvature by Opa1 can influence supramolecular assembly of ATP synthase. We however were prompted by the reviewer to test the very exciting possibility that Opa1 interacts with the ATP synthase to promote its oligomerization. To this end, we took a two-pronged approach and we confirmed Opa1-ATP synthase interaction by co-IP, crosslinking, complexomic analysis of BNPAGE; and excluded, using recombinant OPA1 and purified ATP synthase in vitro in liposome systems, that OPA1 directly promotes oligomerization of the latter. These experiments can be found in **new** Fig. 5. We hope that the reviewer appreciates the efforts for this in vitro system with purified and recombinant proteins.

There is substantial evidence for an autonomous membrane-shaping activity of ATP synthase oligomers independent of OPA1. As the authors present their data in way that OPA1 appears to be the active part in the relationship with ATP synthase, it will be important to directly test, if alterations in ATP synthase dimer/oligomer levels (e.g. ATP5K knock-down or overexpression) affect OPA1 oligomerization. If this is the case, mutual regulation of both machineries appears more likely than a unidirectional process with OPA1 being the trigger.

We thank the reviewer for this insightful comment. We analyzed the oligomerization status of OPA1 upon downregulation of ATP5K and impaired dimerization of ATPase. As expected, OPA1 oligomers are reduced, but only in wt cells, whereas they are sustained by Opa1 overexpression (**new** Supplementary Fig. 7). Indeed, cristae morphology is not altered by downregulation of ATP5K in Opa1^{tg}. These results are in line with the key role of OPA1 oligomerization as a determinant of cristae shape³.

- Overall the paper is very difficult to read. The authors should better explain their assays and rationales to improve accessibility especially for non-expert readers. In particular, the chapter "Opa1 requires ATPase function to maintain mitochondrial function" (page 8f.) is hard to

follow. What do the authors for example mean by the sentence on the differences between TMRM and mtSypHer assays (lane 164ff.)?

Thank you for the suggestions. We have reorganized and rewrote the manuscript to improve its clarity and readability by a broader audience.

Moreover, it is important to show that the decrease in mtATeam ratios upon addition of Antimycin A is indeed oligomycin-sensitive and may therefore unambiguously be assigned to the proton-pumping (i.e. coupled) ATP synthase under the analyzed conditions.

Good point. As suggested, we confirmed that ATP hydrolysis triggered by Antimycin A is oligomycin-sensitive (**new** Fig. 2i), confirming that the observed results rely indeed on ATP synthase reversal activity.

Page 12/13, lines 272-274: In the last part of the Discussion the authors very rightly elaborate on a possible role of the ATPase inhibitor IF1 in linking OPA1 function to the proton-pumping (reverse) ATP synthase/ATPase activity. This is an extremely interesting aspect (however, I do not really understand the mechanistic explanation that is suggested). Therefore, the informative value of this manuscript would benefit a lot from an experimental analysis of the behavior of IF1 under the relevant conditions with blocked respiratory chain and overexpressed or ablated OPA1.

We thank the reviewer for the suggestion. We did not observe any significant changes in IF₁ abundance between WT and *Opa1*^{tg} MAFs. If anything, we saw less IF₁ immunoreactivity in high molecular weight complexes in *Opa1*^{tg} MAFs (**new** supplementary Fig. 5). However, a thorough analysis of the role of Opa1 and IF₁ crosstalk at different matrix pH represents a whole different project. Prompted by the reviewer's comment, we would like to follow this up in the future.

Further scientific and technical points:

Figure 1b/c: The amounts of large OPA1 oligomers in untreated and cBID-treated mitochondria in Figure 1b look very similar. Although the loss of OPA1 oligomers under these conditions is documented again in Figure 1c (and has been extensively studied in the past), still a representative and clear Western blot image should be shown.

Agree. We changed the representative WB.

Page 5, line 99f.: cBID treatment of mitochondria lowers the amount of ATP synthase dimers as shown by BN-PAGE analysis, which is reflected by the loss of the dimerization-promoting subunit ATP5K. Because the overall amount of ATP5K in the gel is much lower upon cBID treatment: What happens to the protein under these conditions (in vitro treatment with cBID)?

Does it dissociate into smaller protein complexes not visible in Figure 1e? Or is the protein degraded (in vitro)?

We have followed the reviewer's suggestions and analyzed the native composition of *ATP5k* (essential for ATPase dimerization) upon BID treatment, which is now included in the **new** Fig. 4e,f. We did not detect any difference in the smaller protein complexes. Instead, as expected, we observed a reduction in the *ATP5k* immunoreactive bands corresponding to ATPase dimers (Vd), in line with their destabilization by BID.

Figure 1f: How many times was this experiment repeated? Error bars should be shown. Moreover, dimer-to-monomer ratios will be the better unit for the y-axis instead of total amounts of dimer counts.

Experiments shown now in Fig. 4 represent the median values of the spectral counts from 3 independent experiments (i.e., 3 independent mitochondria preparations treated as indicated) that were pooled for MS analysis as also previously published by our group⁴. To compare the spectral counts in different bands, we elected to show their median values, which reflect the most typical value, are not skewed by outlier values and as the reviewer appreciates do not allow the calculation of mean or SD values.

Do the authors have any idea, why the amount of ATP5E (epsilon subunit) in the ATP synthase dimer region of the gel is rather increased?

The epsilon subunit (*ATP5E*) is involved in the biosynthesis of F_1 and in its assembly and incorporation to the subunit c into the rotor. It is therefore possible that BID treatment could increase the spectral counts of *ATP5E* upon ATPase structural collapse. Although not measured here, subunit c and epsilon accumulation are comparable in patients with mutations in the *ATP5E*, further suggesting a disassembly of the holoenzyme, with the epsilon subunit helping to incorporate c-subunits to ATPase⁵.

Figure 2e/f: The relative increase of ATPase activity in the dimer bands in Opa1^{tg} mitochondria is not visible in Figure 2e. Is the quantified difference in Figure 1f significant?

We did not find statistically significant differences in resting, non-stimulated ATPase dimer activities (ATP hydrolysis), in line with our kinetic analyses; therefore, we conclude that OPA1 does not stimulate ATPase activity under normal conditions but only when the electron transport chain is inhibited

Figure 2g: Why are the overall amounts of ATP synthase complexes in the mitochondria so different between the samples shown? (The treatment with cBID and variants are done in vitro with isolated mitochondria.) This makes it again very difficult to appreciate ratio differences from the visual inspection of the Western blot. Again, are the quantified differences in Figure 2h/i significant?

Thank you for this remark. We always calculated ratios of ATPase oligomers/total ATP synthase, a measurement that avoids the confusion caused by potentially different total ATP synthase abundance. The differences were significant. To further address the reviewer's comments, we quantified the ATP synthase total abundance, normalized to CII as a loading control, which revealed no significant differences between Wt and *Opa1^{tg}* cells upon challenge with any of the BID mutants (**new** Fig. 3f, h). We have now included oligomers (Vo) and improved resolution of Fig. 3f: now the reduction of higher order ATP synthase assemblies in Wt samples is much more evident.

Page 6, lane 116 ff. and Figure S1c: Why are the total levels of ATP synthase so much reduced in OPA1-deleted MEFs? There is hardly any ATPase activity detected in the gel regions where ATP synthase monomers and dimers are expected. How was a monomer/dimer ratio calculated for these samples?

The observed differences in ATP synthase protein levels in *Opa1*-deleted cells (as included in **new** Supplementary Fig. 2) were expected to occur upon cristae loss and destabilization of dimers and the holoenzyme itself. HMW complexes destabilization is not exclusive for ATP synthase but common to other IMM proteins, as observed when analyzing RCS in our previous work (see e.g. Figure S4 from Cogliati *et al.*²). We agree with the reviewer that ATP synthase activity was hardly visible in *Opa1^{fix/fix}* cells (still, a faint band in the anti-ATPase blot in Fig. 3c left panel is visible). However, densitometric analysis of the bands through high-contrast densitometry of lanes reveals that there are faint bands there, allowing measurements of activity. We include for the reviewer's perusal the analysis of the gel shown in Fig. 3c left. The reviewer can appreciate that there are faint peaks in the oligomer region of the gel (bottom profile, peaks 1&2 from the left)

Figure 3b: There is a problem with the labeling. I assume the red bars represent *mtSypHer* ratios in *Opa1tg* mitochondria.

The reviewer is right. This figure has been now corrected.

I assume that Figure S4a can be considered as "oligomycin only"-control for the experiments in Figure 5c/d. Such controls are also necessary for Figure 5a/b (*mtSypHer* ratio in WT and *Opa1tg* mitochondria) and for figure 5e/f (*mtSypHer* ratios in WT and OPA1-deleted MEFs).

Thank you for the suggestion. The new Supplementary Fig. 3 now show *mtSypHer* fluorescence ratios for the conditions indicated by the reviewer. Matrix pH is comparable among all conditions upon ATP synthase inhibition with oligomycin. This is in line with a similar proton leak across all the cell lines tested and strengthens that OPA1 uses the reversal ATPase activity to prevent matrix acidification induced by antimycin A.

Page 9/10, lines 203 - 206: The logic of this statement is not clear to me? Why to the results supports this idea? There could be other reasons for the observed effects.

Thank you for the comment. In the absence of ATPase dimers at tips, cristae bind to the inner mitochondrial membrane forming septae and archs⁶. In lines 203-206, we referred to the ability of OPA1 to sustain mitochondrial ultrastructure at least partially when ATP dimers are reduced, counteracting the increase in the ratio CJ/cristae as a readout for the formation of septae and archs (new Fig. 6g). We have now rephrased this in the manuscript for clarity.

Page 10, line 218: This sentence refers to Figure 7f (not 6f as indicated).

We have corrected the reference to the new Fig. 7f. Thank you.

Reviewer #2 (Remarks to the Author):

Overall, this paper is well done and the concept is novel. The take-home message is a bit confusing to the reader, is it OPA1 that affects ATPase dimers directly or is this the result of OPA1 effects on cristae morphology? As the mechanism as to how OPA1 mediates these effects is unclear, the manuscript would be significantly strengthened if this point could be better addressed. Below are my other comments.

We would like to thank the reviewer for the appreciation of our study and the insightful suggestions.

We do agree with the reviewer on the need for further analysis of the interaction between OPA1 and ATPase as a key event to be considered over the conclusions of the work. We have now included a whole new set of experiments (shown in **new** Fig. 5) showing that the two proteins can interact. Because physical does not necessarily mean functional interaction, we ran the extra mile and used a reductionist approach of recombinant Opa1, purified ATP synthase and liposomes to verify if Opa1 can per se stimulate ATP synthase oligomerization. Our results show that Opa1 interacts with and stabilizes ATP synthase but is not sufficient to prime its oligomerization in a model membrane.

In figure 2, the authors should provide data to demonstrate that knockout OPA1 does not affect mtDNA or translation. This is mentioned but should also be shown.

We agree with the reviewer that mtDNA levels could interfere with the interpretation of results. However, we have extensively evaluated mtDNA levels and translation in the same *Opa1^{tg}* or *Opa1^{Fix/Fix}* cells used here in a previous publication²: levels and translation of mtDNA are not altered in these model cell lines. We hope that the reviewer agrees that we shall not repeat experiments that we already published on the same exact cell lines.

Many conditions used disrupt cristae and OPA1 together, thus it is unclear whether the effects shown are directly due to OPA1, or secondary effects due to loss of cristae structure in the absence of OPA1. Could the authors find a way to disrupt cristae without altering OPA1? What would happen to the ATPsynthase in that case?

This would be a fantastic experiment. However, in our hands every time that we genetically (e.g., by silencing MICOS components) or apoptotically change cristae ultrastructure, Opa1 oligomers are disrupted^{2, 3 4}. Even if we silence ATP5k we observe that Opa1 oligomers are disassembled in WT cells (**new** Supplementary Fig s7). To overcome these issues, we moved to the reconstituted system described

above that supports a crucial role for IMM shape in favoring ATP synthase supramolecular assembly.

In figure 3 the authors perform a series of complex experiments using matrix pH as a parameter of mitochondrial function. It would also be useful to look at mitochondrial ATP production and oxygen consumption, given that this paper is linking this to ATPsynthase.

We thank the reviewer for the suggestions. By combining TMRM and SypHer experiments, we covered both the electrical and the chemical component of the electrochemical gradient. This is already a step forward compared to studies where TMRM or oxygen consumption are used as proxies of mitochondrial function. Oxygen in the cell models used here has already been extensively addressed previously by us^{2,7}, showing that loss of OPA1 oligomers leading to cristae widening result in compromised respiration.

Unfortunately, we can't use oxygen consumption or ATP production here, because we poison the respiratory chain with AA and we would not expect to detect any oxygen consumption or ATP production. Rather, we can follow mitochondrial matrix ATP levels, which we did in Figs 2, 7 and supplementary Fig. 8.

In figure 4 and 5 the authors discuss how OPA1 mediated protection and maintenance of mitochondria function requires ATPase activity. The authors should show what happens to OPA1 HMWC (high molecular weight complexes) in the absence of ATP synthase activity?

Thank you for the suggestion. We analyzed the native composition of ATPase in isolated mitochondria treated with oligomycin to block ATPase activity. As now shown in **new** Supplementary Fig. 4, we did not observe any change in Opa1 HMWC when the ATP synthase was blocked.

Why are there no changes in basal ATP content in OPA1tg (Figure 5) when in Figure 2 it shows that there is increased ATPase activity in OPA1tg? This should be explained.

Thank you for the comment. As expected, we did not observe any significant change in matrix ATP levels induced by OPA1 overexpression (Supplementary Fig. 3d), probably reflecting that mitochondrial ATP is exported out of the organelle. We now explain this in the revised text

There should be more discussion on how OPA1 affects ATPsynthase dimerization. Is it by OPA1 directly or is it indirectly through OPA1-mediated effects on cristae architecture. As the mechanism by which this occurs is unknown, the authors should try to clarify this point.

We thank the reviewer for the suggestion. We now experimentally address this in the **new** Fig. 5. Our conclusion is that OPA1 promotes ATP synthase oligomerization by its effect on cristae morphogenesis rather than via direct interaction.

Reviewer #3 (Remarks to the Author):

Quintana-Cabrera et al., has investigated the protective role of OPA1 against apoptosis induced mitochondrial dysfunction. Using a variety of BID mutants which dissect the process of apoptosis into outer membrane permeabilization and/or cristae remodelling, Quintana-Cabrera demonstrates that:

- 1) OPA1 is important for the stability of ATP synthase complexes*
- 2) OPA1 overexpression protects against BID induced ATP synthase destabilisation,*
- 3) OPA1 overexpression prevents matrix acidification and membrane potential loss caused by complex III inhibition. (A process requiring active ATP synthase dimers)*
- 4) OPA1 overexpression counteracts changes in cristae morphology caused by reduction in concentration of ATP synthase dimers.*
- 5) Cytochrome c release requires permeabilisation of the outer mitochondrial membrane but not cristae remodelling.*

From these results, the authors predict that OPA1 protects against apoptosis by promoting ATP synthase dimerization and 'reversal activity' (e.g. ATP hydrolysis).

We would like to thank the reviewer for the detailed revision of the manuscript and extensive suggestions to improve the quality of the manuscript.

Main comments:

The manuscript in its current state is inaccessible to the general readership of Nature communications. The authors have a strong tendency to state a hypothesis, reference a figure and then draw a conclusion without properly describing the results being presented. This means the reader needs to be an expert in the techniques used by the authors in order to understand what has lead the authors to draw their conclusions. In addition, because the main figure legends lack sufficient details, it is impossible to understand the figures without the text or the text without the figures.

Thank you for the suggestions. We fully agree and we completely rewrote the manuscript. We hope that the manuscript reads much better now.

Another major issue with the manuscript is the significance of the results discussed. The authors do not discuss the degree of change required to make a result significant. E.g. How significant is a 5nm change in cristae width given the pixel size used for imaging and the technique used (Fig 2b)?

The reviewer is rightfully concerned with the significance of the results we describe and puts forward 3 examples. We will discuss them one by one and we hope to convince the reviewer that all the effects reported in the manuscript are significant. First, the reviewer is concerned that our combination of EM and detector is not able to resolve objects that are separated e.g. by the 5.5 nm difference reported here for

cristae width (28.7 WT-23.2 *Opa1^{tg}*). In TEM, theoretical resolution calculated by combining Abbe and De Broglie equations and by solving then for $\alpha=10^{-2}$ radians and 100kV (the velocity at which we acquire images in our FEI Tecnai G2 microscope) is 0.24nm. We use a Veleta CCD camera as detector, with a detector size of 2048 x 2048 pixels (13 μ m x 13 μ m) and we image at a TEM magnification of 49,000X, with post magnification of 3X. This means that each pixel on the images analyzed has a 0.1x0.1 nm size. As the reviewer correctly points out, in osmicated samples resolution is not limited by the microscope or camera, but by the specimen preparation method. Because the grain size of the metal fixate is ~2nm we have therefore changed the units of the measurements in Fig. 6d and Supplementary 1d to pixels. In both cases, a statistically significant difference was described by accurate morphometry on multiple samples (performed in a blinded fashion).

Is a difference of 0.006 a.u./min really a substantial change (Fig 5i)?

The reviewer is concerned that ATPase activity measured as FRET changes of the ATeaM indicator and expressed as a.u. are not substantial. However, WT are 36% slower than *Opa1^{tg}* cells at hydrolyzing ATP (WT: 0.225 vs. *Opa1^{tg}*:0.6216). We consider a 1/3 difference substantial and indeed, this is statistically significant.

Is a difference of 1 spectral count significant or in the realms of noise (Fig 1f)?

We think that we mistakenly might have ingenerated this confusion. In Fig.1f (now Fig. 5), we plot median values_of spectral counts. Therefore, a value of 1 does not mean that there is a change in 1 spectral count, but that the median differs by 1. Median is much more robust than mean, less influenced by outliers and applicable to non-normally distributed series of data. Thus, changes in 1 median spectral counts is not large, but is significant. The reviewer can appreciate changes in absolute spectral counts from the contour color coded plots of Fig. 4d, e.

In the principle conclusions of the manuscript the authors state that OPA1 promotes “reversal activity” of the ATP synthase. More concrete evidence is required to support this claim, and also evidence to indicate the effect of OPA1 overexpression on the IF1 protein whose function is to prevent ATP hydrolysis by the ATP synthase in mammalian cells. In the authors discussion, they suggest OPA1 interferes with the function of IF1 but the two proteins are in different compartments of the mitochondria. How would this work?

We agree with the reviewer that IF₁ may be a key player to consider in the bioenergetic conservation exerted by OPA1. We evaluated whether levels of IF₁ are affected by Opa1 overexpression and found no significant differences (**new** Supplementary Fig. 5). We think that IF₁ binding to ATP synthase, which peaks at

pH<6.5⁹, might be delayed in *Opa1*^{tg} cells treated with AA by the prevented matrix acidification (shown here). Thus, by promoting ATP synthase oligomerization and reversal activity, Opa1 prevents matrix acidification and interrupts also the feedback loop involving IF₁ inactivation of the ATP synthase. Future work should also evaluate the reciprocal impact of IF₁ on Opa1 processing¹⁰ under the experimental conditions reported here.

In addition, it is not clear that OPA1 promotes ATP synthase dimerization from the data presented. I agree that the authors present clear data demonstrating that active ATP synthase dimers are required for overexpressed OPA1 to protect against matrix acidification and membrane potential loss caused by complex III inhibition but there is no evidence to say that it actively promotes ATP synthase dimerization or how this might occur.

The reviewer is right, and we took this comment very seriously. We ran the extra mile to verify how Opa1 might promote higher order assembly of ATP synthase. First, we demonstrated using a variety of approaches that Opa1 and ATP synthase are retrieved in the same HMW complexes and that subunits of the ATP synthase can physically interact with Opa1. Because this is no proof of functional interaction, we employed an in vitro system of artificial membranes, recombinant Opa1 and purified ATP synthase to test if Opa1 was per se able to promote oligomerization of the ATP synthase, but this was not the case (Fig. 5g, h). We therefore performed a set of biochemical and genetics experiments that show that ATP synthase oligomerization and cristae shape is stabilized by Opa1 (Figs 3, 5, 6), suggesting that these higher order assemblies of the ATP synthase are favored by the cristae curvature generated by Opa1. We think that these further experiments substantiate a model whereby ATP synthase oligomerization is indeed promoted by structural components of the complex, but also by cristae curvature. This extends a key concept in membrane biology proposed a few years ago by us², whereby changes in membrane shape can influence reactions carried out by membrane-embedded macromolecular complexes by modulating supramolecular assemblies of the latter.

Point-by-point comments

Why use ATPase to refer to the mitochondrial ATP synthase. In cells, the main function of the ATPase is ATP synthesis not hydrolysis (hence ATP synthase).

Apologies, we did so for the sake of brevity. We now refer to the enzyme as ATP synthase

Title: The data presented demonstrates that the protective effect of OPA1 on mitochondrial function requires active ATP synthase dimers. This should be reflected more accurately in

the title. E.g. '... requires active mitochondria ATP synthase dimers' instead of '... requires mitochondrial ATP synthase dimerization'

Thank you for the suggestion. Title is now modified

Abstract: I do not believe the data presented provides sufficient evidence to support the notion that OPA1 stimulates ATPase dimerization. In addition, the data present does not show apoptotic and genetic manipulation based changes to cristae shape. This is inferred from previous publication and should be removed from the abstract. Instead the abstract should be rewritten to focus on the concrete results of the paper as stated in the title.

We hope that the new version of the abstract reads better.

Introduction

Line 42: Ref 11 and 12 are duplications

Thank you for pointing this out. We have now corrected the references.

Line 77-79: I do not think the data presented provides sufficient reliable evidence to support the statement "OPA1 promotes ATPase dimerization and reversal activity". Also I did not encounter a clear explanation or presentation of the mechanisms through "which cristae stabilization by OPA1 protects from primary mitochondrial dysfunction". In addition, what do the authors mean by "primary mitochondrial dysfunction."?

We have rewritten the paper and we hope that now the text adheres much better to the presented results.

Results section "ATPase dimers are destabilized during cristae remodelling"

Line 86: Purified heart mitochondria was obtained from which animal? Wild type / overexpression?

Samples were obtained from wild type mice (CD1 strain). This is now specified in the Methods section ⁴

Line 91: therefrom should be changed to henceforth

Done. Thank you.

Line 98: "Fig. 1c) revealed that cristae remodelling resulted in the expected disassembly of HMW OPA1-containing complexes". There seems to be a discrepancy between the complexome and western blot images. In fig 1b, OPA1 appears to been present at roughly the same level as in the untreated mitochondria but in the complexome (fig 1c) there appears to be a large difference. This needs to be explained.

We have now included a more representative blot and inset in Fig. 4b showing OPA1 destabilization by BID. Please also note that complexome analysis is more sensitive than WB.

Line 95: Why use the gene name for the subunits of the ATP synthase when you are detecting the protein?

Thank you for the comment. We have used the gene name provided from the identification of peptides through the database. We aimed at avoiding confusions by using both the gene names and ATP synthase protein subunit name in parenthesis.

Line 96-97: “were reduced during cristae remodelling in the BNGE regions corresponding to ATPase dimers (Fig 1d)”. The region on the gel corresponding to the ATPase dimers need to be indicated for the reader.

Done.

Line 102: “ the spectral counts remained ~72% of the untreated (Fig 1e)” and Fig 1F. The graph for subunit e shows a difference of one spectral count between BID and BID^{KKAA} treated mitochondria. When is a change in spectral count considered significant? Why does the spectral count for ATP5E triple in the BID treated mitochondria?

Data refer to median spectral count values, and a change of one unit is dependent on the series of values and less susceptible to outliers.

Why does the spectral count stay the same between BID and BID^{KKAA} treated mitochondria for the ATP5L? The spectral counts for ATP5L and ATP5E are in the same range as ATP5K. How reliable are these results and how much can you conclude from these data?

Different accessibility to digitonin extraction may be responsible for these differences, which are also due to the folding status of ATPase or differences in the role of these subunits in the biosynthesis or stability of ATPase.

For instance, the epsilon subunit (*ATP5E*), participates in the biosynthesis of the F₁ part and its assembly and incorporation to the subunit c into the rotor⁵. *ATP5L* (subunit *g*), also involved in ATPase dimerization, is enriched in untreated mitochondria, whereas it has similar median spectral counts in *BID* and *BID^{KKAA}* challenged mitochondria. This may be a result of *ATP5L* loss during partial destabilization of the subunit upon outer membrane permeabilization or a higher retention of the subunit by the holoenzyme also after *BID* treatment.

Results section “ATPase stability is regulated by Opa1 levels”

Line 110: more explanation as to why each stage is performed would make the manuscript easier to read. E.g. "Because OPA1 overexpression stabilizes mitochondrial RCS, we verified if the same was also true for the ATPase. For these experiments we used adult mouse fibroblasts (MAFs) derived from transgenic mice which were overexpressing OPA1 (Opa1^{tg}). In these cells, OPA1 was expressed 1.5-fold greater than the WT and had narrower gaps in the cristae. ..."

Thank you for the comment. We have completely rewritten the manuscript to take all your suggestions into account

Line 112 & Fig 2A: How was the 1.5-fold increase in Opa1 quantified?

OPA1 protein levels were quantified by densitometry (Image J, NIH) and reflect the mRNA levels expressed in Opa1^{tg} MAFs, as in our previous works^{2,7}.

Line 112 & Fig 2B: How significant is the 5 nm narrowing of the cristae especially given the pixel size, camera resolution and fixation procedure? How many different mitochondria were analysed from how many cells? Were these from the same fixation experiment or different? Could the change in distance reflect different angled slices through the lamellar cristae or even different amounts of dehydration between different cells during the fixation/embedding procedure?

We counted cristae width in all the visible cristae (on average 10 per mitochondrion) in at least 5 mitochondria/cell in at least 6 randomly selected cells per condition, all repeated in 3 different, independent experiments. This means that we measured at least 900 cristae widths per point of our morphometric analysis. This description is now clarified in the "methods" section. Experimental conditions and processing of samples remained the same among different preparations, which were run in parallel with their respective controls to avoid interferences of processing and image acquisition. As for the plane orientation, we of course never compared orthogonal slices (which are a source of artefact especially in specialized tissues like the heart or skeletal muscle), but always acquired images of comparably oriented sections.

Line 133 & Fig 2d-f: What evidence leads you to the conclusion that OPA1 over expression increases the stability of the F1Fo ATP synthase dimer? Why is figure S1c in the supplementary?

Reviewer is right. Now this image is in the main figures (**new** Fig. 3a, b) and we strengthen the conclusion by including a densitometric analysis of the levels of oligomeric/total ATP synthase.

Line 115: How do you know that you didn't alter mtDNA content or translation caused by chronically impaired fusion? Could mtDNA content or translation be modified other than by chronically impaired fusion? Also what exactly does 'chronically impaired fusion' mean? Are you referring to cell-cell fusion or mitochondrial fusion?

mtDNA depletion/translation can be affected by chronic Opa1 deletion. This was addressed in our previous work², where we demonstrated that chronically impaired mitochondrial fusion due to the genetic deletion of mitofusins (*Mfn*^{-/-}) or Opa1 (*Opa1*^{-/-}) cells results in mtDNA loss and mitochondrial protein abundance. Conversely, mild Opa1 overexpression in *Opa1*^{tg} cells or acute deletion of the protein in *Opa1*^{fix/fix} cells (upon Cre recombinase expression) does not affect mtDNA load and translation.

Line 124: "cBID induced complete cytochrome c release in WT but not Opa1^{tg} MAFs" What indicates complete release?

BID challenge in permeabilized cells results in the release of the bulk of cytochrome c retained in the cristae, which represents ~85% of the total stores in mitochondria¹¹. However, these experiments did not add much to the paper and to improve clarity we elected to remove them from the revised version.

Line 126: "(iii) cytochrome c release was comparably reduced when we treated WT cells with cBID^{KKAAA} or Opa1^{tg} cells with cBID". How do you define "comparably reduced"? What are you comparing the results with? Why is this data in the supplementary figures when you discuss it at length in the main text?

This set of data validated that apoptotic cristae remodelling widens the cristae to release cytochrome c and that OPA1 prevents this to occur. This was included in supplementary materials to prove an impact on cytochrome c mobilization, while other key experiments addressed the consequences on bioenergetics in the main figures. Using the elegant assay originally described by Hajnoczky and colleagues¹² that uses decreases in TMRM fluorescence as a proxy for cytochrome c release, we showed a similar cytochrome c release in Wt cells treated with *cBID*^{KKAAA} (which makes pores in the outer membrane but does not cause cristae remodeling) and in *Opa1*^{tg} cells challenged with *cBID* (which permeabilizes the outer membrane and causes cristae remodelling). This indicates that OPA1 overexpression keeps cytochrome c inside the cristae and that the amount of cytochrome c released in both conditions corresponds to the ~15% located in the intermembrane space¹¹. However, as for the previous point, these mutants had been extensively characterized in previous publications and we elected not to show them in the revised version, for the sake of clarity.

Line 127: “Thus, OPA1-controlled cristae remodelling is required for complete cytochrome c release also in permeabilized cells”. Is this conclusion valid? As described by the authors, BID^{KKAA} permeabilises the outer membrane but does not cause cristae remodelling (Line 89) and BID^{G94E} causes cristae remodelling but without permeabilizing the outer mitochondrial membrane (line 121). Fig S2 indicates cytochrome c is released using cBID and BID^{KKAA} but not BID^{G94E} indicating cytochrome c release requires outer membrane permeabilisation but not cristae remodelling.

We apologize if we were not clear enough here. The keyword here is “complete”: release across the outer membrane requires outer membrane permeabilization and the action of the multidomain proapoptotic Bcl-2 family members. Cristae remodeling is conversely necessary for the release of the bulk of cytochrome c from the cristae¹¹. In this sense, the loss of Opa1 oligomers is necessary to widen the cristae and facilitate the exit of the cytochrome c³ through the pores formed upon BID recruitment to the outer membrane. The BID^{G94E} is a control to show that under these experimental settings, cristae remodelling without outer membrane permeabilization and cytochrome c release does not result in membrane potential loss. However, as for the previous point, these mutants had been extensively characterized in previous publications and we elected not to show them in the revised version, for the sake of clarity.

Lines 122-128: Why did the authors choose to place the graphs for this section as supplementary and not main text?

This figure was used to further characterize these cBID mutants (already characterized in several other publications) in the experimental setting used here. However, we elected not to show them in the revised version, for the sake of clarity.

Line 132 & fig 2g-i: Why is the figure labelled with BID and not cBID? What is the difference?

Thank you. cBID refers to recombinant, caspase-8 cleaved BID. To avoid confusion, we use now BID throughout the paper.

Line 133: “Thus in chronically or acutely remodelled cristae ATPase dimers are destabilized” What generates chronically remodelled cristae and what produces acutely remodelled cristae? What is the difference between chronic and acutely remodelled cristae? Do you have data to support these assignments?

These words were used to differentiate the situation where we deleted Opa1 (chronically remodelled) from the one where we treated with BID (acutely remodelled). We agree that this does not sound correct and we now rewrote this section.

The only conclusions that can be drawn from this section is that 1) ATP synthase stability decreases with cristae remodelling and 2) overexpression of OPA1 appears to protect against BID induced ATP synthase destabilisation.

We do agree with the reviewer on the main conclusions drawn at the end of the section. The description of the role of BID at inducing cristae widening and cytochrome c release was intended as validation of our results on bioenergetics. However, as for the previous point, these mutants had been extensively characterized in previous publications and we elected not to show them in the revised version, for the sake of clarity.

Line 133: Does cBID^{G94E} and cBID cause cristae remodelling in Opa1^{tg}? Please provide evidence for this.

This was studied in our previous works ^{2,7}. However, as for the previous points, these mutants had been extensively characterized in previous publications and we elected not to show them in the revised version, for the sake of clarity.

Results section “OPA1 counteracts mitochondrial dysfunction caused by cristae remodelling”

Line 139: “Matrix pH was comparable in WT and Opa1^{tg} MAFs”. How are they comparable?

We refer here to the finding that basal matrix pH measured using SypHer was similar, as in **new** Supplementary Fig 1e. We changed comparable with similar in the text.

Line 146. Corresponding figures need to be referred to at beginning of explanation.

Done.

Line 151: Shouldn't tBID^{G94A} be tBID^{G94E}?

The reviewer is right. Thank you.

Line 153: Δ pH should be replaced with matrix acidification as this is what the authors are measuring.

Done.

Line 153: “induced by the tBID mutants that cause cristae remodelling”. Only one mutant affects cristae remodelling (tBID^{G94E}). Also the protection offered by OPA1 overexpression seems to be equally strong in all tBID variants not just the one that affects cristae remodelling (fig 3a). The authors must provide addition proof that the protection offered by OPA1 overexpression has more effect on the mutant which only remodels the cristae.

Apologies for having caused this confusion. Both BID^{G94E} and BID WT widen the cristae, with the difference that the former does not cause outer membrane permeabilization. Since BID is not a mutant, we have now corrected the text. The experiments in Fig. 3 showed that AA-induced matrix acidification is stronger when cristae are remodelled by BID WT or BID^{G94E} and that these effects are prevented in *Opa1*^{tg} cells. On the contrary, BID^{KKAA} that does not remodel cristae does not aggravate matrix acidification. However, as for the previous points, these mutants had been extensively characterized in previous publications and we elected not to show them in the revised version, for the sake of clarity.

Fig 3b +AA. What change in ratio indicates a significant difference? Is a 0.05 change in ratio really significant? What does the "ratio" measure?

SypHer is a ratiometric probe that has been extensively characterized by Santo-Domingo *et al.*¹³. Ratios indicate the ratio between the grey values (i.e., the fluorescence intensity) recorded in the regions of interest (ROI) of the images acquired at 535 nm emission upon alternate excitation of the sample at 430 and 500 nm. Considering the ratiometric nature of the probe, a 0.05 ratio change (430/500) means that the ROI values for the image recorded exciting at 500nm were 20 times higher than those in the same ROI for the image recorded exciting at 430nm. However, as for the previous point, these mutants had been extensively characterized in previous publications and we elected not to show them in the revised version, for the sake of clarity.

Line 154: conclusion should represent the conditions tested. E.g. "OPA1 overexpression protects cells against matrix acidification caused by inhibition of complex III"

The section title should also be modified to reflect the data discussed e.g. OPA1 overexpression protects against matrix acidification caused by complex III inhibition.

Thank you for the suggestions. We have now rephrased this.

Results section "Opa1 requires ATPase activity to maintain mitochondrial function"

Line 163: What does "fully protected" mean?

We mean that OPA1 overexpression abolishes significant TMRM fluorescence variations upon AA treatment.

Line 169: "Mitochondrial dysfunction" is a broad term. Authors should clarify which type of "mitochondrial dysfunction" they are referring to?

Thank you for the comment. We have now rephrased it.

Line 172: Can the authors really say that OPA1 requires mitochondrial ATPase activity to sustain ΔpH upon CIII inhibition? Isn't it more likely that oligomycin inhibition of ATP synthase over-rides (is dominant to) complex III inhibition by Antimycin A? If ATP synthase function is inhibited, the respiratory chain complexes also become inhibited due to a lack of dissipation of the intra-cristae proton concentration. The authors need to discuss this. A better conclusion would be: "the protection OPA1 overexpression provides on matrix acidification caused by complex III inhibition requires active ATP synthases"

We rephrased as suggested. However, we would like to point out that AA already inhibits the respiratory chain (RC): by inhibiting e⁻ transfer at complex III, e⁻ can't flow downhill through redox couples of the RC and H⁺ pumping at complex I and IV does not occur, as validated by a plethora of bioenergetics studies and as the reviewer perfectly knows. Thus, oligomycin in the presence of AA does not increase ΔpH : if anything, it contributes to its faster dissipation because of the inhibition of the ATP synthase reversal (see e. g. Fig. 2).

Line 175: The ATPase activity of F-type ATP synthases in cells (especially mammals) is minimal because as the matrix acidifies and the conditions become appropriate for ATP hydrolysis by the ATP synthase, the inhibitor factor (IF1) binds to the ATP synthase preventing ATP hydrolysis.

The reviewer is right. Binding of IF₁ occurs under conditions where a pH close to 6.5 is reached in the matrix⁹, therefore preventing the ATP hydrolysis by the ATPase and preserving ATP when mitochondria are uncoupled. ATPase is active at hydrolyzing ATP until reaching the optimal conditions for IF₁ binding that in turn avoids further ATP depletion in the cell. By avoiding acidification, OPA1 overexpression may thus delay the binding of IF₁ to ATPase. We now discuss this.

Line 178: Oligomycin binds to the membrane embedded rotor ring of ATP synthase preventing the movement of protons across the membrane. Thus in oligomycin inhibited cells there is no loss in membrane potential or ΔpH . This paragraph should be removed or modified to reflect this knowledge.

Of course: this is a basic tenet of bioenergetics that holds true when mitochondria are fully coupled and there is no proton leak. We clarified this.

Line 185: The function of the IF1 protein in mammalian mitochondria is to inhibit ATP hydrolysis by ATP synthase when the matrix becomes acidic. How does OPA1 overexpression and OPA1 knockdowns effect matrix IF1 concentration?

Levels of IF₁ are not affected in Opa1¹⁹ cells (**new** supplementary Fig. 5)

Line 185 & Fig 5g-i. How significant is a change of 0.06 a.u./min (fig 5i) or a mtATeam ratio change of 1% (Fig 5h)?

These measurements are meant to address the speed of ATP hydrolysis, which is reflected by changes in mtATeam slopes, plotted as a.u. Because mtATeam is a FRET probe, it reports the ratio of the 525/475 nm fluorescence emission variations and changes of 0.1 indicate that the denominator (475 nm fluorescence) is 10-fold higher than the 525 nm fluorescence numerator. These differences are better shown by measuring the slope of ATP hydrolysis in new Fig. 7e: the first derivative of mtATeam fluorescence changes over time indicate that mtATeam fluorescence decay is 2.5 fold faster in *Opa1^{tg}* cells. Moreover, this increase is fully sensitive to oligomycin and therefore due to ATP synthase activity.

Line 186: "In conclusion, OPA1 stimulates F₁Fo-ATP synthase reversal activity to sustain mitochondrial ΔpH". The authors do not provide any evidence to support this claim. They at least need to validate the effect of OPA1 on matrix IF₁ concentration and the significance of a 0.006 a.u./min change in mtATeam rate.

Thank you for this comment. Several lines of evidence indicate that ATP synthase reversal activity is required for Opa1 to sustain *mitochondrial ΔpH*. First, IF₁ levels are unaffected by Opa1 overexpression (supplementary fig. 7); second, mtATeam fluorescence decay (fully sensitive to oligomycin and therefore due to ATP synthase activity) is 2.5-fold faster in *Opa1^{tg}* cells.

Line 194: Please change ATPase to ATPase.

Done. Thank you.

Line 199: "While ATP5k downregulation did not affect cristae width". This is not possible to accurately measure with the technique used.

We hope that this was clarified in our response above. With more than 900 morphometric measurements, we believe that we convinced the reviewer that this parameter was not affected by this genetic manipulation, whereas it was by Opa1 deletion or upregulation. Because of the valid point on the nm conversion of the pixel differences we measured, we changed the measurement units of width into pixels.

Line 219: Please change "blunt" to curtail.

Done.

Results section "OPA1 requires ATPase dimerization to maintain mitochondrial function"

Line 189: “In Opa1⁹”, this should be changed to “In mitochondria from Opa1 overexpression MAFs”.

Done.

Line 189: “cristae are narrower” This is difficult to quantify or validate with the technique used.

Please see above (line 199)

Line 189: “ATPase is more dimerized and active”. The data presented in the manuscript provides no evidence for more dimers only that they are more stable (less likely to disassemble)

Thank you. We have rephrased this sentence to: “ATPase oligomers are more abundant”

Line 191: “mitochondrial dysfunction is a too broader term. Please be more specific by describing the type of mitochondrial dysfunction you have analysed or are going to test.

Thank you for the comment. We have now substituted it by “mitochondria CIII blockage”

Discussion

Line 230, 235: It is not clear what the authors means by “ameliorates” and “amelioration”. Please exchange for a simpler word. This would make the statements more accessible to readers where English is not their first language.

Done.

Line 272: My understanding is that OPA1 is an intra-crista space protein which in some form binds to the inner mitochondrial membrane. IF1 is a soluble matrix protein. How can overexpression of OPA1 prevent binding of the IF1 to the ATP synthase when they are not located in the same compartment?

Thank you for the comment. The modulation of mitochondrial bioenergetics and matrix pH would be key for Opa1 to indirectly delay IF₁ binding to ATPase, which occurs at acidic pH⁹. Another indirect relationship is also feasible, since IF1 could prevent Opa1 cleavage by activation of OMA1 upon depolarization¹⁰. Finally, the interaction of OPA1 with ATPase, as now demonstrated by our co-IP and crosslink experiments, could work as a bridge to IF₁.

Figures:

Figure 1:

1) *Please indicate location of ATP synthase dimer and monomer band. Done*

- 2) *Please complete parenthesis in figure 1F.* Done.
- 3) *What does (--) mean? Please add to figure legend description.* We have substituted – by “vehicle”.
- 4) *Why are figures 1a-e labelled cBID and cBID^{KKAA} and fig. 1F is labelled BID^{WT} and BID^{KKAA}?*
What is the difference. They are the same and are now corrected. Thank you.
- 5) *What does the color number scale refer to?* They refer to the number of spectral counts. We have added this to the figure legend.
- 6) *What does the numbers on the left and right (fig 1c) refer to?* They refer to the excision bands for mass spectroscopy, now indicated in the figure legend.

Supplementary figure 1:

- 1) *What is the staining used in the left image of S1C?* It is a BNGE-PAGE blot immunoblotted against ATPA and anti-SDHA. We have now indicated this in the figure legend.
- 2) *Fig S1C, left label ATPA should be changed to ATPase. ATPA refers to ATPase subunit alpha (F1 sector).* Done.

Figure 2:

- 1) *Fig 1c, y-axis + bars should have a break to indicate scale does not extend from zero (see Fig 1h).* Corrected. Thank you.
- 2) *Fig 1d, please change the label ATPA to ATPase to make it consistent with text. Also add label to figure legend to improve clarity.* Done. We have also detailed this information in the figure legend for clarity.

Figures 3-7:

- 1) *What do the color bar mean in the first fluorescent image of each panel? What is the meaning of the scale that is indicated?* The pseudocolor bar indicates the fluorescence correspondence to SypHer ratios. We have specified this in the figure legends.
- 2) *AA, EV and FCCP need to be defined in the figure legend.* Done.
- 3) *shSCR and shATP5k need to be explained in figure legend.* Done.
- 4) *What is being tested in each image?* This is specified in the figure legends (i.e. mtATeam SypHer or TMRM fluorescence, electron microscopy images). Done.

Methods:

Line 364: F0. 0 should be o as it stands for oligomycin. Done. Thank you.

References

1. Patten,D.A. *et al.* OPA1-dependent cristae modulation is essential for cellular adaptation to metabolic demand. *EMBO J.* **33**, 2676-2691 (2014).
2. Cogliati,S. *et al.* Mitochondrial cristae shape determines respiratory chain supercomplexes assembly and respiratory efficiency. *Cell* **155**, 160-171 (2013).
3. Frezza,C. *et al.* OPA1 controls apoptotic cristae remodeling independently from mitochondrial fusion. *Cell* **126**, 177-189 (2006).
4. Glytsou,C. *et al.* Optic Atrophy 1 Is Epistatic to the Core MICOS Component MIC60 in Mitochondrial Cristae Shape Control. *Cell Rep.* **17**, 3024-3034 (2016).
5. Mayr,J.A. *et al.* Mitochondrial ATP synthase deficiency due to a mutation in the ATP5E gene for the F1 epsilon subunit. *Hum. Mol. Genet.* **19**, 3430-3439 (2010).
6. Habersetzer,J. *et al.* Human F1F0 ATP synthase, mitochondrial ultrastructure and OXPHOS impairment: a (super-)complex matter?. *PLoS. One.* **8**, e75429 (2013).
7. Varanita,T. *et al.* The OPA1-dependent mitochondrial cristae remodeling pathway controls atrophic, apoptotic, and ischemic tissue damage. *Cell Metab* **21**, 834-844 (2015).
8. Bammes,B.E., Rochat,R.H., Jakana,J., Chen,D.H., & Chiu,W. Direct electron detection yields cryo-EM reconstructions at resolutions beyond 3/4 Nyquist frequency. *J. Struct. Biol.* **177**, 589-601 (2012).
9. Cabezon,E., Butler,P.J., Runswick,M.J., & Walker,J.E. Modulation of the oligomerization state of the bovine F1-ATPase inhibitor protein, IF1, by pH. *J. Biol. Chem.* **275**, 25460-25464 (2000).
10. Faccenda,D. *et al.* Control of Mitochondrial Remodeling by the ATPase Inhibitory Factor 1 Unveils a Pro-survival Relay via OPA1. *Cell Rep.* **18**, 1869-1883 (2017).
11. Scorrano,L. *et al.* A distinct pathway remodels mitochondrial cristae and mobilizes cytochrome c during apoptosis. *Dev. Cell* **2**, 55-67 (2002).
12. Roy,S.S. & Hajnoczky,G. Fluorometric methods for detection of mitochondrial membrane permeabilization in apoptosis. *Methods Mol. Biol.* **559**, 173-190 (2009).
13. Santo-Domingo,J., Giacomello,M., Poburko,D., Scorrano,L., & Demareux,N. OPA1 promotes pH flashes that spread between contiguous mitochondria without matrix protein exchange. *EMBO J.* **32**, 1927-1940 (2013).

REVIEWERS' COMMENTS:

Reviewer #1 (Remarks to the Author):

In the (substantially) revised version of this manuscript the authors have carefully and comprehensively addressed all major points raised by the reviewers. Moreover, with its new structure and organization the manuscript will be much better accessible, particularly for non-expert readers. The new data largely confirm the original conclusions of the authors.

Reviewer #2 (Remarks to the Author):

The authors have address all of my concerns, I am happy with the revisions. This paper addresses a very important question and will be of interest to a broad audience. I enthusiastically support publication.

Reviewer #3 (Remarks to the Author):

Quintana-Cabrera et al have submitted an excellent revision of their manuscript titled "The cristae modulator Optic atrophy 1 requires activity of mitochondrial ATP synthase oligomers to safeguard mitochondrial function". Compared to their first version, the manuscript is very clear, presents some interesting and thought-provoking experiments and was a joy to read. The contents is now very accessible to the general readership of Nature Communications and will be an excellent contribution to the journal.

However, before publishing, the authors need to address some issues that have arisen in the new version of the manuscript. The most important issues are located on page 10.

Page 5: line 125. "Cre recombinase delivery", please can the authors provide a brief explanation of what this is for the uninitiated reader.

Page 7 line 159-160. The wording of this sentence is confusing: Are the authors trying to say that when the ATP synthase inhibitor oligomycin disrupted OP1 oligomerisation, the changes in fluorescence were not due to ATP synthase activity?

Page 7 line 161: How do the authors know OPA1 overexpression "stimulates" reversal ATP synthase activity? It is probably more accurate to say that the protection provided by OPA1 overexpression involves or is linked to rates of ATP hydrolysis.

Page 8 line 181-185. If BIDKKA does not alter cristae morphology and does not change ATP synthase oligomerization patterns, does this mutant provide similar protection to mitochondria as OPA1 overexpression?

Page 9 line 209: Wording: how do the authors know the stabilization of ATP synthase oligomers are stabilized by OPA1 overexpression? It is probably more accurate to exchange

the word "by" with "when".

Page 9 line 210: Wording. It is not clear whether you need both cristae remodeling and OPA1 HMW oligomers to disassemble to destabilized the ATP synthase oligomers or whether one is sufficient.

Page 9 line 217: Reference 44 investigates the oligomeric state of the ATP synthase inhibitory factor (IF1). The authors of ref44 did not investigate or provide any evidence to suggest IF1 preserves ATP synthase dimerization upon matrix acidification. The sentence on line 217 needs to be modified to accurately reflect the experiments of ref 44.

Page 10 line 229: "with their F1 heads facing outside, as confirmed by EM (fig 5d)". Fig 5d is a micrograph of liposomes stained with uranyl acetate (e.g. negative staining). The Uranyl acetate coats the outside of vesicles and thus it is impossible to say which direction the ATP synthases have been reconstituted as only those with their catalytic domains on the outside would be detected using this method. Thus Fig 5d confirms that the ATP synthases have been incorporated into vesicles but not the orientation.

Page 10 line 236: "since contaminant protein with ATPase activity co-purified with rOPA1, we removed it by extensive washes with ATP". Please can the authors explain the chemistry behind this observation? If the proteins have bound the column based on charge complement to the Ni-NTA group on the bead, how does the ATPase activity of the contaminant and washing the column with ATP break this interaction?

Page 10 line 241: "We therefore incorporated rOPA1 into the proteoliposome lumen, mimicking the relative Topology of OPA1 and ATP synthase in mitochondria". LOGIC. On page 9, line 221, the authors say they detect a cross-link between the beta-subunit and OPA1. The beta-subunit of ATP synthase is located in the matrix, 5nm from the cristae membrane. OPA1 has a single transmembrane helix for membrane insertion and the majority of the protein is exposed to the lumen of the cristae. As these two proteins are located on opposite sides of the membrane, is the cross-link observed between the beta subunit and OPA1 biologically significant? Also if you wanted to recreate that cross-link, you need to locate both proteins in the same compartment.

Page 11: The experiment reported on page 11 is very interesting. Does OPA1 overexpression increase the lifetime (reduce the turnover rate) of ATP synthase? This might explain why there are more ATP synthase oligomers around in OPA1 overexpression and why, given the same levels of IF1 you get ATP hydrolysis during the inhibition of complex III with AA in the OPA1 overexpression mutants.

Page 12: line 289. Probably more accurate to say "Our results indicate that the protection provided by OPA1 overexpression requires ATP5K and efficient ATP synthase oligomerization to sustain mitochondrial Δ pH and curtail cell death following complex III Inhibition.

Line 298: An explanation of what the authors mean by reversal activity probably needs to be provided somewhere in the manuscript.

Line 299: What are multiple insults? Can the authors be more specific?

Line 324: Do the authors mean 'latter' rather than 'latest'?

FIGURES and FIGURE LEGEND

The abbreviations that oligo=oligomycin and AA=Antimycin A need to be stated in the figures legends. Also the abbreviations EV, Cre and Vehicle need to be explained.

Figure 4: What type of cells were used in this experiment? Were they wild type for OPA1 or OPA1 overexpression mutants? What is the significance of short and long exposure for figure 4b? which was used for the spectral analysis?

Supplementary figure 6. The label surnatant could be replaced by supernatant as there is enough room.

Response to reviewers' comments

Reviewer #1 (Remarks to the Author):

In the (substantially) revised version of this manuscript the authors have carefully and comprehensively addressed all major points raised by the reviewers. Moreover, with its new structure and organization the manuscript will be much better accessible, particularly for non-expert readers. The new data largely confirm the original conclusions of the authors.

We thank the reviewer for the appreciation of our revision

Reviewer #2 (Remarks to the Author):

The authors have address all of my concerns, I am happy with the revisions. This paper addresses a very important question and will be of interest to a broad audience. I enthusiastically support publication.

We thank the reviewer for the appreciation of our revision

Reviewer #3 (Remarks to the Author):

Quintana-Cabrera et al have submitted an excellent revision of their manuscript titled "The cristae modulator Optic atrophy 1 requires activity of mitochondrial ATP synthase oligomers to safeguard mitochondrial function". Compared to their first version, the manuscript is very clear, presents some interesting and thought-provoking experiments and was a joy to read. The contents is now very accessible to the general readership of Nature Communications and will be an excellent contribution to the journal.

We thank the reviewer for the appreciation of our revision

However, before publishing, the authors need to address some issues that have arisen in the new version of the manuscript. The most important issues are located on page 10.

We thank the reviewer for the useful comments

Page 5: line 125. "Cre recombinase delivery", please can the authors provide a brief explanation of what this is for the uninitiated reader.

We changed it into the clearer "after transfection with CRE recombinase"

Page 7 line 159-160. The wording of this sentence is confusing: Are the authors trying to say that when the ATP synthase inhibitor oligomycin disrupted OPA1 oligomerisation, the changes in fluorescence were not due to ATP synthase activity?

No, we meant that oligomycin does not affect OPA1 oligomerization, and that the changes in fluorescence are due to ATP synthase activity. The whole sentence now reads "These fluorescence changes were indeed due to ATP synthase activity, because they were abolished by the ATP synthase inhibitor oligomycin (Fig. 2g-i). Notably, oligomycin did not affect OPA1 oligomerization in WT or Opa1tg MAFs (Supplementary Fig. 4)."

Page 7 line 161: How do the authors know OPA1 overexpression "stimulates" reversal ATP synthase activity? It is probably more accurate to say that the protection provided by OPA1 overexpression involves or is linked to rates of ATP hydrolysis.

Because the ATP hydrolysis stimulated by Opa1 is inhibited by oligomycin, indicating that it is mediated by the ATP synthase.

Page 8 line 181-185. If BIDKAA does not alter cristae morphology and does not change ATP synthase oligomerization patterns, does this mutant provide similar protection to mitochondria as OPA1 overexpression?

This mutant was characterized in Cogliati et al, Cell 2013 and as the reviewer knows we had a set of confirmatory experiments in the first version of the manuscript that were expunged for the sake of clarity in this version: similar amounts of cytochrome c are released from WT mitochondria treated with this BID mutant and from Opa1tg mitochondria treated with WT BID.

Page 9 line 209: Wording: how do the authors know the stabilization of ATP synthase oligomers are stabilized by OPA1 overexpression? It is probably more accurate to exchange the word "by" with "when".

Done. We changed by with upon

Page 9 line 210: Wording. It is not clear whether you need both cristae remodeling and OPA1 HMW oligomers to disassemble to destabilized the ATP synthase oligomers or whether one is sufficient.

Opa1 oligomer disassembly accompanies cristae remodeling. The sentence now reads "or cristae remodeled with concomitant OPA1 HMW oligomers disassembly"

Page 9 line 217: Reference 44 investigates the oligomeric state of the ATP synthase inhibitory factor (IF1). The authors of ref44 did not investigate or provide any evidence to suggest IF1 preserves ATP synthase dimerization upon matrix acidification. The sentence on line 217 needs to be modified to accurately reflect the experiments of ref 44.

Apologies. We restructured the sentence and refer also to the paper by Fernando Minauro-Sanmiguel et al. PNAS 2005 that suggests a role for IF oligomers in stabilizing ATPase dimers. The sentence now reads “whose oligomerization is influenced by matrix pH4.4 and that in its oligomeric form can stabilize ATP synthase dimers⁴⁵”

Page 10 line 229: “with their F1 heads facing outside, as confirmed by EM (fig 5d)”. Fig 5d is a micrograph of liposomes stained with uranyl acetate (e.g. negative staining). The Uranyl acetate coats the outside of vesicles and thus it is impossible to say which direction the ATP synthases have been reconstituted as only those with their catalytic domains on the outside would be detected using this method. Thus Fig 5d confirms that the ATP synthases have been incorporated into vesicles but not the orientation.

Agree. We deleted “with their F1 heads facing outside” from the sentence

Page 10 line 236: “since contaminant protein with ATPase activity co-purified with rOPA1, we removed it by extensive washes with ATP”. Please can the authors explain the chemistry behind this observation? If the proteins have bound the column based on charge complement to the Ni-NTA group on the bead, how does the ATPase activity of the contaminant and washing the column with ATP break this interaction?

The contaminant protein was bound to recombinant Opa1, as it was not binding to the Ni-NTA column in the absence of rOpa1. The washes with ATP allowed probably a conformational change that dissociated the contaminant from rOpa1 and eluted it. We state now that this contaminant “bound” rather than “co-purified”

Page 10 line 241: “We therefore incorporated rOPA1 into the proteoliposome lumen, mimicking the relative Topology of OPA1 and ATP synthase in mitochondria”. LOGIC. On page 9, line 221, the authors say they detect a cross-link between the beta-subunit and OPA1. The beta-subunit of ATP synthase is located in the matrix, 5nm from the cristae membrane. OPA1 has a single transmembrane helix for membrane insertion and the majority of the protein is exposed to the lumen of the cristae. As these two proteins are located on opposite sides of the membrane, is the cross-link observed between the beta subunit and OPA1 biologically significant? Also if you wanted to recreate that cross-link, you need to locate both proteins in the same compartment.

The reviewer is right. This in vitro experiment was exactly meant to go beyond the consensus that colP or crosslinking are enough to establish biologically meaningful interaction. When two proteins located in different compartments, as the reviewer correctly points out, are shown to interact, a more careful analysis like the one shown here is required. We therefore leave the cross-linking experiment in, and the reader to understand that, perhaps, many of the reported “interactions” are flawed by the fact that the proteins are not localized in the right compartments or properly oriented.

Page 11: The experiment reported on page 11 is very interesting. Does OPA1 overexpression increase the lifetime (reduce the turnover rate) of ATP synthase? This might explain why there are more ATP synthase oligomers around in OPA1 overexpression and why, given the same levels of IF1 you get ATP hydrolysis during the inhibition of complex III with AA in the OPA1 overexpression mutants.

We thank the reviewer for this insightful suggestion, which we will follow up in future studies.

Page 12: line 289. Probably more accurate to say "Our results indicate that the protection provided by OPA1 overexpression requires ATP5K and efficient ATP synthase oligomerization to sustain mitochondrial ΔpH and curtail cell death following complex III Inhibition.

Done

Line 298: An explanation of what the authors mean by reversal activity probably needs to be provided somewhere in the manuscript.

Done, we say "(i.e., ATP hydrolase)" in that sentence

Line 299: What are multiple insults? Can the authors be more specific?

Here, we used insults in its medical meaning, i.e. the cause of some physical damage like the thermal, electrical or radiation cause of a skin burn- we added descriptors for the medically illiterate reader.

Line 324: Do the authors mean 'latter' rather than 'latest'?

Yes. We changed it, thanks for pointing it out

FIGURES and FIGURE LEGEND

The abbreviations that oligo=oligomycin and AA=Antimycin A need to be stated in the figures legends. Also the abbreviations EV, Cre and Vehicle need to be explained.

Done

Figure 4: What type of cells were used in this experiment? Were they wild type for OPA1 or OPA1 overexpression mutants? What is the significance of short and long exposure for figure 4b? which was used for the spectral analysis?

We used mouse heart mitochondria, as stated in the figure legend. This is an in vitro experiment. The long exposure highlights the 720kDa OPA1 complexes. The spectral analysis is

not performed on immunoblots, but by proteomic analysis on complexes excised from the BN-PAGE, as described in the figure legend

Supplementary figure 6. The label supernatant could be replaced by supernatant as there is enough room.

Done